# Conformal Frequency Estimation with Sketched Data

**Matteo Sesia**
Department of Data Sciences and Operations
University of Southern California
Los Angeles, California, USA
sesia@marshall.usc.edu

**Stefano Favaro**
Department of Economics and Statistics
University of Torino and Collegio Carlo Alberto
Torino, Italy
stefano.favaro@unito.it

## Abstract

A flexible conformal inference method is developed to construct confidence intervals for the frequencies of queried objects in very large data sets, based on a much smaller sketch of those data. The approach is data-adaptive and requires no knowledge of the data distribution or of the details of the sketching algorithm; instead, it constructs provably valid frequentist confidence intervals under the sole assumption of data exchangeability. Although our solution is broadly applicable, this paper focuses on applications involving the count-min sketch algorithm and a non-linear variation thereof. The performance is compared to that of frequentist and Bayesian alternatives through simulations and experiments with data sets of SARS-CoV-2 DNA sequences and classic English literature.

## 1 Introduction

### 1.1 Frequency queries from sketched data

An important task in computer science is to estimate the frequency of an object given a lossy compressed representation, or *sketch*, of a big data set [1, 2]; this task has real-world relevance in diverse fields including machine learning [3], cybersecurity [4], natural language processing [5], genetics [6], and privacy [7]. Practically, sketching may be motivated for example by memory limitations, as large numbers of distinct symbols may otherwise be computationally expensive to analyze [6], or by privacy constraints, in situations where the original data contain sensitive information [8]. There exist many sketching algorithms, several of which are specifically designed to enable efficient approximations of the empirical frequencies of the compressed objects. We refer to the monograph of [9] for a recent review of sketching. Building upon this literature, our paper studies the problem of precisely quantifying the uncertainty of empirical frequency estimates obtained from sketched data without making strong assumptions about the inner workings of the sketching procedure, which may be complex and even unknown. The key ideas of the described solution are in principle applicable regardless of how the data are compressed, but the exposition of this paper will focus for simplicity on a particularly well-known sketching algorithm and some widely applied variations thereof.

### 1.2 The count-min sketch

The count-min sketch (CMS) of [10] is a renowned algorithm for compressing a data set of $m$ objects $Z_1, \ldots, Z_m \in \mathcal{Z}$, with $\mathcal{Z}$ being a discrete (and possibly infinite) space, into a representation with reduced memory footprint, while allowing simple approximate queries about the observed frequency of any possible $z \in \mathcal{Z}$. At the heart of the CMS lie $d \geq 1$ different $w$-wide *hash* functions $h_j : \mathcal{Z} \to [w] := \{1, \ldots, w\}$, for all $j \in [d] := \{1, \ldots, d\}$ and some integer $w \geq 1$. Each hash function maps the elements of $\mathcal{Z}$ into one of $w$ possible buckets, and it is designed to ensure that distinct values of $z$ populate the buckets uniformly. Hash functions are typically chosen at random from a *pairwise independent* family $\mathcal{H}$, which ensures the probability (over the randomness in the

choice of hash functions) that two distinct objects $z_1, z_2 \in \mathcal{Z}$ are mapped by two different hash functions into the same bucket is equal to $1/w^2$. The data set $Z_1, \ldots, Z_m$ is then compressed into a sketch matrix $C \in \mathbb{N}^{d \times w}$ with row sums equal to $m$. The element in the $j$-th row and $k$-th column of $C$ counts the number of data points mapped by the $j$-th hash function into the $k$-th bucket:

$$C_{j,k} = \sum_{i=1}^{m} \mathbb{1}\left[h_j(Z_i) = k\right], \qquad j \in [d]. \tag{1}$$

Because $d$ and $w$ are such that $d \cdot w \ll m$, the matrix $C$ loses information compared to the full data set, but it has the advantage of requiring much less space to store.

Given a sketch $C$ from (1), one may want to answer queries about the original data. A fundamental instance of this problem consists in estimating the true empirical frequency of an object $z \in \mathcal{Z}$:

$$f_m(z) := \sum_{i=1}^{m} \mathbb{1}\left[Z_i = z\right]. \tag{2}$$

One solution is to return the smallest count among the $d$ buckets into which $z$ is mapped:

$$\hat{f}_{\mathrm{up}}^{\mathrm{CMS}}(z) = \min_{j \in [d]}\{C_{j,h_j(z)}\} \geq f_m(z). \tag{3}$$

This procedure is outlined in Algorithm A1 (Appendix A1) and it provides a deterministic upper bound for $f_m(z)$ [10]. Of course, $\hat{f}_{\mathrm{up}}^{\mathrm{CMS}}(z)$ tends to overestimate $f_m(z)$ due to possible hash collisions. But the independence of the hash functions ensures collisions are not too common, leading to a classical probabilistic lower bound for $f_m(z)$.

**Theorem 1** ([10]). *Fix any $\delta, \epsilon \in (0, 1)$. Suppose $d = \lceil -\log \delta \rceil$ and $w = \lceil e/\epsilon \rceil$. Then, $\mathbb{P}_{\mathcal{H}}[f_m(z) \geq \hat{f}_{\mathrm{up}}^{\mathrm{CMS}}(z) - \epsilon m] \geq 1 - \delta$ for any fixed $z \in \mathcal{Z}$, where $\hat{f}_{\mathrm{up}}^{\mathrm{CMS}}(z)$ is as in (1) and depends on $d, w$.*

For example, if $d = 3$, this says that a 95% lower bound on $f_m(z)$ is $\hat{f}_{\mathrm{up}}^{\mathrm{CMS}}(z) - \lceil e \cdot m/w \rceil$. The subscript $\mathcal{H}$ in Theorem 1 clarifies that the randomness is with respect to the hash functions, while $Z_1, \ldots, Z_m$ and $z$ are fixed. The above bound can be useful to inform the choices of $d$ and $w$ prior to sketching, but it is not fully satisfactory as a way of quantifying the uncertainty about the true frequency of a given query. First, it is often too conservative [11] if the data are randomly sampled from some distribution as opposed to being fixed. Second, it is not flexible: $\delta$ cannot be chosen by the user because it is fixed by $d$, and $\epsilon$ is uniquely determined by the hash width. Thus, in general Theorem 1 cannot give a reasonably tight confidence interval for $f_m(z)$ at an arbitrary level. As we will discuss below, these limitations can be overcome by assuming the data are sampled randomly.

### 1.3 Uncertainty estimation for sketching with random data

An alternative approach to computing probabilistic lower and upper bounds for $f_m(z)$ was proposed by [11] in order to address the often excessive conservativeness of the classical bounds described above. The approach of [11] is based on bootstrapping ideas and departs from classical analysis of the CMS as it leverages randomness in the data instead of randomness in the hash functions. Precisely, [11] assumes the data and the queried object are independent and identically distributed (i.i.d.) random samples from some unknown distribution. These assumptions may not always be justified in practice, but they are useful because they can lead to much more informative confidence intervals. In fact, the confidence intervals described by [11] are nearly exact and optimal for the CMS, up to a possible finite-sample discrepancy between the bootstrap and population distributions.

A limitation of the bootstrap method of [11] is that it relies on the specific *linear* structure of the CMS—the sketch matrix $C$ in (3) can be written as a linear combination of the true frequencies of all objects in the data set—and thus the idea is not easily extendable to other sketching algorithms that may perform better than the CMS in practice. One issue with the CMS is that it is relatively sensitive to random hash collisions, which can result in overly conservative deterministic upper bounds. This challenge has motivated the development of alternative *non-linear* sketching algorithms, such as the CMS with *conservative updates* (CMS-CU). Whenever a new object $Z_i$ is sketched, the CMS-CU only updates the row of $C$ with the smallest value of $C_{j,h_j(z)}$, leaving the other counters unaltered. Then, the same CMS deterministic upper bound in (3) remains valid. This procedure is outlined in Algorithm A2. This non-linear approach can lead to much higher query accuracy compared to the vanilla CMS [12], but the theoretical analysis of the CMS-CU beyond a deterministic upper bound is more challenging, and it appears to be a relatively less explored topic.

## 1.4 Contributions

This paper develops a novel method for constructing valid and reasonably tight frequentist confidence intervals for frequency queries based on sketched data. Our method works seamlessly with the CMS-CU as well as with any possibly unknown sketching algorithm. The solution assumes the data and the query are random samples from some unknown population, making only minimal exchangeability assumptions about this data generating process. Although our method requires higher computational power and memory usage compared to pure sketching, the additional cost of our uncertainty estimates will be negligible compared to the typical size of the data sets involved.

## 1.5 Related work

Different types of lower bounds have been developed for the CMS [9], although most treat the data as fixed and only utilize the hashing randomness, similarly to Theorem 1. More recently, uncertainty estimation for the CMS based on data randomness has been studied by [11] from a frequentist perspective, as mentioned in Section 1.3, and by [13, 14] from a Bayesian perspective. More precisely, [13] and [14] model the data with a prior distribution and compute the posterior of the queried frequencies given the sketch. Our work is closer to [11] in that we treat the data as random while seeking frequentist probabilistic guarantees. However, our approach is very different from that of [11]: the latter exploits the specific linear structure of the CMS, while we view the sketch as a black box and utilize conformal inference to obtain exact finite-sample inferences.

Conformal inference was pioneered by Vovk and collaborators [15] and brought to the statistics spotlight by [16]. Although primarily conceived for supervised prediction [17–21], conformal inference has found many other applications including outlier detection [22], causal inference [23], and survival analysis [24]. To the best of our knowledge, its potential for sketching remained untapped until now. There exist many algorithms for using sketches to compute approximate frequency queries; some are similar to the CMS [9, 25–27], while others are more complicated and may involve complex learning algorithms [28, 29]. In theory our method is applicable to all of them, but here we focus more on the CMS and variations thereof because it is a classic algorithm and it provides a familiar example that facilitates the exposition. As we will make use of conformal inference, a brief review of the relevant technical background is provided in the next section.

## 2 Preliminaries on conformal prediction

Conformal prediction is typically concerned with *supervised learning*, in which the data samples are pairs $(X_i, Y_i)$, with $X_i$ indicating a vector of *features* for the $i$-th observation and $Y_i$ denoting the corresponding continuous (or discrete) *label*. In supervised learning, the goal is to use the information contained in $(X_1, Y_1), \ldots, (X_m, Y_m)$ to learn a mapping between features and labels that can allow one to predict as accurately as possible the unseen label $Y_{m+1}$ of a new sample with features $X_{m+1}$. In particular, conformal prediction assumes that $(X_1, Y_1), \ldots, (X_{m+1}, Y_{m+1})$ are exchangeable random samples from some unknown joint distribution over $(X, Y)$ and then constructs a prediction interval $[\hat{L}_{m,\alpha}(X_{m+1}), \hat{U}_{m,\alpha}(X_{m+1})]$ with guaranteed *marginal coverage*:

$$\mathbb{P}[\hat{L}_{m,\alpha}(X_{m+1}) \leq Y_{n+1} \leq \hat{U}_{m,\alpha}(X_{m+1})] \geq 1 - \alpha, \tag{4}$$

for any fixed $\alpha \in (0, 1)$. Conformal prediction is flexible, as it can leverage any machine learning algorithm to approximately reconstruct the relation between $X$ and $Y$, thus yielding relatively short intervals satisfying (4). Note that, while it is appropriate to focus on conformal intervals in this paper, similar techniques can also be utilized to construct more general prediction sets [30, 31].

The simplest version of conformal prediction begins by randomly splitting the available observations into two disjoint subsets, assumed for simplicity to have equal size $n = m/2$. The first $n$ samples are spent to fit a black-box machine learning model for predicting $Y$ given $X$; e.g., a neural network or a random forest. The out-of-sample predictive accuracy of this model is then measured in terms of a *conformity score* for each of the $n$ hold-out data points. In combination with the model learnt from the first half of the data, the empirical distribution of these scores is translated into a recipe for constructing prediction intervals for future test points as a function of $X_{m+1}$; these intervals are guaranteed to cover $Y_{n+1}$ with probability at least $1 - \alpha$, treating all data as random. The details of this procedure will be clarified shortly. Importantly, the coverage holds exactly in finite samples,

regardless of the accuracy of the machine learning model, as long as $X_{m+1}$ is exchangeable with the $n$ hold-out data points. Note that it is unnecessary for the training data to be also exchangeable, as these may be seen as fixed, but it is sometimes easier to say all data points are exchangeable.

The implementation of conformal inference depends on the choice of conformity scores. While several different options exist, an intuitive and general explanation is the following. Imagine that associated with the fitted machine learning model there exists a *nested sequence* [32] of prediction intervals $[\hat{L}_{m,\alpha}(x;t), \hat{U}_{m,\alpha}(x;t)]$ for each possible $x$. This sequence is indexed by $t \in \mathcal{T} \subseteq \mathbb{R}$, which may be either discrete or continuous, and it is increasing: $\hat{L}_{m,\alpha}(x;t_2) \leq \hat{L}_{m,\alpha}(x;t_1)$ and $\hat{U}_{m,\alpha}(x;t_2) \geq \hat{U}_{m,\alpha}(x;t_1)$ for all $t_2 \geq t_1$. Further, assume there exists $t_\infty \in \mathcal{T}$ such that $\hat{L}_{m,\alpha}(x;t_\infty) \leq Y \leq \hat{U}_{m,\alpha}(x;t_\infty)$ almost surely for all $x$. For example, one may consider a sequence $\hat{\psi}_m(x) \pm t$, where $\hat{\psi}_m$ is a regression function for a bounded label $Y$ given $X$ learnt by the black-box machine learning model and $t$ plays the role of a predictive standard error. Then, the conformity score for a data point with features $X = x$ and label $Y = y$ can be defined as the smallest index $t$ such that $y$ is contained in the sequence of prediction intervals corresponding to $x$:

$$E(X, Y) := \min \left\{ t \in \mathcal{T} : Y \in [\hat{L}_{m,\alpha}(x;t), \hat{U}_{m,\alpha}(x;t)] \right\}. \tag{5}$$

The conformal prediction rule is then easily stated. Let $\mathcal{I}^{\text{calib}} \subset \{1, \ldots, m\}$ be the subset of hold-out data points, assumed without loss of generality to have size $n = m/2$. Let $\hat{Q}_{n,1-\alpha}$ be the $\lceil (1-\alpha)(n+1) \rceil$ smallest value among the $n$ conformity scores $E(X_i, Y_i)$ evaluated for all $i \in \mathcal{I}^{\text{calib}}$. The conformal prediction interval for a new data point with features $X_{m+1}$ is:

$$\left[ \hat{L}_{m,\alpha}(X_{m+1}; \hat{Q}_{n,1-\alpha}), \hat{U}_{m,\alpha}(X_{m+1}; \hat{Q}_{n,1-\alpha}) \right]. \tag{6}$$

Intuitively, this satisfies (4) because $Y_{m+1}$ is outside (6) if and only if $E(X_{m+1}, Y_{m+1}) > \hat{Q}_{n,1-\alpha}$. The rest of the proof is a simple exchangeability argument; see [20] or the proof of Theorem 2.

## 3 Conformalized sketching

### 3.1 Exchangeable queries and marginal coverage

Assume the $m$ data points to be sketched, $Z_1, \ldots, Z_m \in \mathcal{Z}$, are exchangeable random samples (a weaker condition than i.i.d.) from some distribution $P_Z$ on $\mathcal{Z}$. Although $P_Z$ may be arbitrary and unknown, our framework is more restrictive compared to the classical setting reviewed in Section 1.2, which treats the data as fixed and thus can also handle non-stationary streams or adversarial cases. Yet, imagining the data as approximately i.i.d. samples from some distribution is not an unprecedented idea in the context of sketching [11, 13, 14], and it may be justified when objects from a large data set are processed in a random order; see Sections 4 and 5. As we shall see below, data exchangeability can be an extremely useful assumption because it allows one to obtain powerful inferences while taking a completely agnostic view of the inner workings of the sketching algorithm.

Consider an arbitrary *sketching* function $\phi : [\mathcal{Z}]^m \to \mathcal{C}$, where $\mathcal{C}$ is a space with lower dimensions compared to $[\mathcal{Z}]^m$. For example, $\phi$ may represent the sketching performed by the CMS or CMS-CU, in which case $\mathcal{C} = \mathbb{N}^{d \times w}$, for some $d, w$ such that $d \cdot w \ll m$. In general, we will treat $\phi$ as a *black box* and allow it to be anything, possibly involving random hashing or any other data compression technique. The goal is to leverage the data exchangeability and the information in $\phi(Z_1, \ldots, Z_m)$ to estimate the unobserved true empirical frequency $f_m(z)$ of some object $z \in \mathcal{Z}$, defined as in (2), while accounting for uncertainty. More precisely, although still informally speaking, we would like to construct a *confidence interval* $[\hat{L}_{m,\alpha}(z), \hat{U}_{m,\alpha}(z)]$ guaranteed to contain $f_m(z)$ "with probability at least $1 - \alpha$", where the randomness here is with respect to the random data sampling.

One way to address the above problem is to imagine the query $z$ is also randomly sampled exchangeably with $Z_1, \ldots, Z_m$. This is a convenient but quite strong assumption that will be partly relaxed later. With this premise, we refer to $z$ as $Z_{m+1}$ and focus on computing a confidence interval $[\hat{L}_{m,\alpha}(Z_{m+1}), \hat{U}_{m,\alpha}(Z_{m+1})]$ depending on $\phi(Z_1, \ldots, Z_m)$ that is reasonably short and guarantees marginal coverage of the true unknown frequency:

$$\mathbb{P}\left[ \hat{L}_{m,\alpha}(Z_{m+1}) \leq f_m(Z_{m+1}) \leq \hat{U}_{m,\alpha}(Z_{m+1}) \right] \geq 1 - \alpha, \tag{7}$$

where the probability is with respect to the data in $Z_1, \ldots, Z_m$ as well as to the randomness in the query $Z_{m+1}$. The interpretation of (7) is as follows: the interval will cover the true frequency $f_m(Z)$ for at least a fraction $1 - \alpha$ of all future test points $Z$ on average, if the queries and the sketched data are re-sampled exchangeably. In Section 3.3, we will develop a method for constructing reasonably tight confidence intervals satisfying (7) exactly. This is already a non-trivial result on its own, but marginal coverage is not fully satisfactory because some objects may be queried more often than others, and therefore not all of our *distinct* inferences are equally reliable. In particular, the confidence intervals for rare queries may have lower coverage, as illustrated by the following thought experiment. Imagine $P_Z$ has support on $\mathcal{Z} = \{0, 1, 2, \ldots, 10^{100}\}$, with $\mathbb{P}[Z_i = 0] = 0.95$ and $\mathbb{P}[Z_i = z] = 0.05/(|\mathcal{Z}| - 1)$ for all $z \in \mathcal{Z} \setminus \{0\}$ and $i \geq 1$. Then, a 95% confidence interval satisfying (7) may be completely unreliable for all but one specific query about $f_m(0)$.

## 3.2 Beyond full exchangeability with approximate frequency-conditional coverage

To begin addressing the limitations of marginal coverage, Section 3.3 will develop a more refined method for constructing confidence intervals that are simultaneously valid for both rare and common random queries. Our approach re-purposes relevant ideas from Mondrian conformal inference [33], which have been utilized before to construct prediction sets with label-conditional coverage for classification problems [15, 30, 34]. However, departing from the typical approach in classification, we will not seek perfect coverage conditional on the exact true frequency of the queried object. In fact, that problem would be impossible to solve without stronger assumptions [35], because $f_m(Z_{m+1})$ can take a very large number of possible values when the sketched data set is big. Instead, we will focus on achieving a relaxed concept of frequency-conditional coverage which groups together different queries of objects that appear a similar number of times within the sketched data set. Precisely, let us fix any partition $\mathcal{B} = (B_1, \ldots, B_L)$ of $\{1, \ldots, m\}$ into $L$ bins such that each $B_l$ is a sub-interval of $\{1, \ldots, m\}$, for some relatively small integer $L$. In theory, this partition should be fixed prior to seeing the data and may not depend on the new query $Z_{m+1}$. Our goal is to construct a confidence interval $[\hat{L}_{m,\alpha}(Z_{m+1}), \hat{U}_{m,\alpha}(Z_{m+1})]$ depending on $\phi(Z_1, \ldots, Z_m)$ and $\mathcal{B}$ that is reasonably short and satisfies the following notion of *frequency-range conditional coverage*:

$$\mathbb{P}\big[\hat{L}_{m,\alpha}(Z_{m+1}) \leq f_m(Z_{m+1}) \leq \hat{U}_{m,\alpha}(Z_{m+1}) \mid f_m(Z_{m+1}) \in B\big] \geq 1 - \alpha, \qquad \forall B \in \mathcal{B}. \quad (8)$$

The choice of $\mathcal{B}$ involves some trade-offs: finer partitions (larger $L$) yield more reliable theoretical guarantees but possibly wider confidence intervals; in particular, intervals associated with a bin $B$ may tend to be wider if $\mathbb{P}[Z \in B]$ is small. In practice, we will adopt $|\mathcal{B}| = 5$ later in this paper, but much larger $|\mathcal{B}|$ can be utilized when working with very big data, as it will become clear below.

## 3.3 Conformalization methodolody

In the attempt of adapting conformal inference to address our sketching problem, the first difficulty is that the latter is a data recovery problem, not a supervised prediction task. We propose to overcome this challenge by storing in memory the true frequencies for all objects in the first $m_0$ observations, with $m_0 \ll m$ but sufficiently large subject to memory constraints. Without loss of generality, assume $m_0 \ll m$; otherwise, the problem becomes trivially easy. Let $n \leq m_0$ indicate the number of distinct objects among the first $m_0$ observations. The memory required to store these frequencies is $O(n)$, which will typically be negligible as long as $m_0$ is also small compared to the size of the sketch, $|\mathcal{C}|$. This approach turns our problem into a supervised prediction one, as detailed below.

During an initial *warm-up* phase, the frequencies of the $n$ distinct objects among the first $m_0$ observations from the data stream, $Z_1, \ldots, Z_{m_0}$, are stored exactly into:

$$f_{m_0}^{\mathrm{wu}}(z) := \sum_{i=0}^{m_0} \mathbb{1}[Z_i = z]. \quad (9)$$

Next, the remaining $m - m_0$ data points are streamed and sketched, storing also the true frequencies for all instances of objects already seen during the warm-up phase. In other words, the following counters are computed and stored along with the sketch $\phi(Z_{m_0+1}, \ldots, Z_m)$:

$$f_{m-m_0}^{\mathrm{sv}}(z) := \begin{cases} \sum_{i=m_0+1}^{m} \mathbb{1}[Z_i = z], & \text{if } f_{m_0}^{\mathrm{wu}}(z) > 0, \\ 0, & \text{otherwise.} \end{cases} \quad (10)$$

Again, this requires only $O(n)$ memory. Now, let us define the variable $Y_i$ for all $i \in \{1, \ldots, m_0\} \cup \{m + 1\}$ as the true frequency of $Z_i$ among $Z_{m_0+1}, \ldots, Z_m$:

$$Y_i := \sum_{i'=m_0+1}^{m} \mathbb{1}\left[Z_{i'} = Z_i\right]. \tag{11}$$

Note that $Y_i$ is observable for $i \in \{1, \ldots, m_0\}$, in which case $Y_i = f_{m-m_0}^{\mathrm{sv}}(Z_i)$. For a new query, $Z_{m+1}$ is the target of inference—in truth, the target is $f_m(Z_i) = Y_i + f_{m_0}^{\mathrm{wu}}(Z_i)$, but the second term is already known exactly. To complete the connection between sketching and supervised conformal prediction, one still needs to define meaningful features $X$, and this is where the sketch $\phi(Z_{m_0+1}, \ldots, Z_m)$ comes into play. For each $i \in \{1, \ldots, m_0\} \cup \{m + 1, \ldots\}$, define $X_i$ as the vector containing the object of the query as well as all the information in the sketch:

$$X_i := (Z_i, \phi(Z_{m_0+1}, \ldots, Z_m)). \tag{12}$$

The following result establishes that the pairs $(X_1, Y_1), \ldots, (X_{m_0}, Y_{m_0}), (X_{m+1}, Y_{m+1})$ are exchangeable with one another. This anticipates that conformal prediction techniques can be applied to the supervised observations $(X_1, Y_1), \ldots, (X_{m_0}, Y_{m_0})$ to predict $Y_{m+1}$ given $X_{m+1}$, guaranteeing the marginal coverage property in (7). All mathematical proofs are in Appendix A2.

**Proposition 1.** *If the data $Z_1, \ldots, Z_{m+1}$ are exchangeable, then the pairs of random variables $(X_1, Y_1), \ldots, (X_{m_0}, Y_{m_0}), (X_{m+1}, Y_{m+1})$ defined in* (11)–(12) *are exchangeable with one another.*

Confidence intervals satisfying frequency-range coverage (8) can be obtained by modifying the standard conformal inference procedure as outlined in Algorithm A3, whose details are deferred to Appendix A1 for lack of space. First, each point $(X_i, Y_i)$ in $\mathcal{I}^{\mathrm{calib}} = \{1, \ldots, m_0\}$ is assigned to the appropriate frequency bin based on $Y_i$. Define $n_l$ as the number of calibration points in bin $B_l$, for all $l \in \{1, \ldots, L\}$. Then, $\hat{Q}_{n_l, 1-\alpha}(B_l)$ is defined as the $\lceil (1 - \alpha)(n_l + 1) \rceil$ smallest value among the $n_l$ scores assigned to bin $B_l$. Finally, the confidence interval for a new random query $X_{m+1}$ is:

$$\left[\hat{L}_{m,\alpha}(X_{m+1}; \hat{Q}_{n,1-\alpha}^*), \hat{U}_{m,\alpha}(X_{m+1}; \hat{Q}_{n,1-\alpha}^*)\right],$$

where

$$\hat{Q}_{n,1-\alpha}^* := \max_{l \in \{1,\ldots,L\}} \hat{Q}_{n_l,1-\alpha}(B_l),$$

and $[\hat{L}_{m,\alpha}(\cdot; t), \hat{U}_{m,\alpha}(\cdot; t)]$ is a rule for computing a nested sequence of intervals depending on $Z_{m+1}$ and $\phi(Z_{m_0+1}, \ldots, Z_m)$, assumed by convention to be increasing in $t$. Examples of such rules are in the next section. Importantly, these rules may involve parameters to be fitted on a subset of $m_0^{\mathrm{train}}$ supervised data points $(X_i, Y_i)$ for $i \in \{1, \ldots, m_0^{\mathrm{train}}\}$, as long as the scores are only evaluated on the remaining $m_0 - m_0^{\mathrm{train}}$ points. The following result states that the confidence interval output by Algorithm A3 has the desired frequency-range conditional coverage.

**Theorem 2.** *If the data $Z_1, \ldots, Z_{m+1}$ are exchangeable, the confidence interval output by Algorithm A3 satisfies the frequency-range conditional coverage property defined in* (8).

Theorem 2 also implies that confidence intervals satisfying the weaker marginal coverage property defined in (7) can be obtained by applying Algorithm A3 with the trivial partition $\mathcal{B}$ dividing the range of possible frequencies into a single bin of size $m$, because (7) is a special case of (8). Note also that Algorithm A3 could be trivially modified to output perfect "singleton" confidence intervals for any new queries that happen to be identical to an object previously observed during the warm-up phase. We will not take advantage of this fact in the experiments presented in this paper to provide a fairer comparison with alternative methods which do not involve a similar warm-up phase.

### 3.4 Conformity scores for one-sided confidence intervals

Algorithm A3 can accommodate virtually any data-adaptive rule for computing nested confidence intervals, which may depend on $Z_{m+1}$ and on the sketch $\phi := \phi(Z_{m_0+1}, \ldots, Z_m)$. Two concrete options are presented here. For simplicity, we focus on one-sided confidence intervals; that is, we seek a $1 - \alpha$ lower bound for $f_m(X_{m+1})$. This is useful when a deterministic upper bound $\hat{f}_{\mathrm{up}}(Z_{m+1}) \geq f_m(Z_{m+1})$ is already available, as it is the case with the CMS or CMS-CU. Then,

one can simply set $\hat{U}_{m,\alpha}((z,\phi);t) := \hat{f}_{\mathrm{up}}(z)$ and focus on computing $\hat{L}_{m,\alpha}(\cdot;t)$. The construction of two-sided confidence intervals is explained in Appendix A1.4 due to lack of space.

To construct the lower bound $\hat{L}_{m,\alpha}(\cdot;t)$ of a one-sided interval, the first option is to use a fixed rule:

$$\hat{L}_{m,\alpha}^{\mathrm{fixed}}((z,\phi);t) := \max\{0, \hat{f}_{\mathrm{up}}(Z_{m+1}) - t\}, \qquad t \in \{0,1,\ldots,m\}. \tag{13}$$

In words, the lower bound for $f_m(Z_{m+1})$ in (14) is defined by shifting the deterministic upper bound downward by a constant $t$. The appropriate value of $t$ guaranteeing the desired coverage for future random queries is calibrated by Algorithm A3. This approach is intuitive, and it is very similar to the optimal solution of [11] for the special case of the CMS. Further, it does not need training data, so all $m_0$ observations with tracked frequencies can be utilized for computing conformity scores.

The second option involves training but has the advantage of being more flexible; this is inspired by the methods of [36, 37] for regression. Concretely, consider a machine learning model that takes as input the known upper bound $\hat{f}_{\mathrm{up}}(Z_i)$ and estimates the conditional distribution of $\hat{f}_{\mathrm{up}}(Z_i) - f_m(Z_i)$ given $\hat{f}_{\mathrm{up}}(Z_i)$. For example, think of a multiple quantile neural network [38] or a quantile random forest [39]. After fitting this model on the $m_0^{\mathrm{train}}$ supervised data points $(X_i, Y_i)$ allocated for training, let $\hat{q}_t$ be the estimated $\alpha_t$ lower quantile of the of $\hat{f}_{\mathrm{up}}(Z_i) - f_m(Z_i) \mid \hat{f}_{\mathrm{up}}(Z_i)$, for all $t \in [1,\ldots,T]$ and some fixed sequence $1 = \alpha_1 < \ldots < \alpha_T = 0$. Without loss of generality, assume the machine learning model is that $\hat{q}_0 = 0$ and $\hat{q}_T = m$. Then, define $\hat{L}_{m,\alpha}(\circ;t)$ as:

$$\hat{L}_{m,\alpha}^{\mathrm{adaptive}}((z,\phi);t) := \max\left\{0, \hat{f}_{\mathrm{up}}(X_{m+1}) - \hat{q}_t\left(\hat{f}_{\mathrm{up}}(X_{m+1})\right)\right\}, \qquad t \in \{0,1,\ldots,m\}. \tag{14}$$

This second approach can lead to a lower bound whose distance from the upper bound is adaptive. This is an advantage because some sketching algorithms may introduce higher uncertainty about common queries compared to rarer ones, or vice versa, and such patterns can be learnt given sufficient data. Of course, the two above examples of $\hat{L}_{m,\alpha}(\cdot;t)$ are not exhaustive. Algorithm A3 can be applied in combination with any rule for computing nested sequences of lower bounds, and it can leverage all high-dimensional information contained in $\phi(Z_{m_0+1},\ldots,Z_m)$, not just the deterministic upper bound for the CMS and CMS-CU. However, the applications in this paper focus on the CMS and CMS-CU, so other families of lower bounds are not discussed here.

# 4 Applications

## 4.1 Experiments with synthetic data sets

Conformalized sketching is applied with the two types of conformity scores described in Section 3.4, focusing on the construction of one-sided confidence intervals. Additional experiments involving two-sided intervals are described in Appendix A4. The adaptive scores utilized in this section are based on an isotonic distributional regression model [40]. The goal is to compute lower frequency bounds for random queries based on simulated data compressed by the CMS-CU, implemented with $d = 3$ hash functions of width $w = 1000$. In particular, $m = 100,000$ observations are sampled i.i.d. from some distribution specified below. The first $m_0 = 5000$ observations are stored without loss during the warm-up phase, as outlined in Algorithm A3, while the remaining $95,000$ are compressed by the CMS-CU. The conformity scores are evaluated separately within $L = 5$ frequency bins, seeking the frequency-range conditional coverage property defined in (8). The bins are determined in a data-driven fashion so that each contains approximately the same probability mass; in practice, this is achieved by partitioning the range of frequencies for the objects tracked exactly by Algorithm A3 according to the observed empirical quantiles. Lower bounds for new queries are computed for $10,000$ data points also sampled i.i.d. from the same distribution. The quality of these bounds is quantified with two metrics: the mean *length* of the resulting confidence intervals and the coverage the proportion of queries for which the true frequency is correctly covered, or empirical *coverage*. The performance is averaged over 10 independent experiments.

Experiments are performed on synthetic data sampled from two families of distributions. First, we consider a Zipf distribution with $\mathbb{P}[Z_i = z] = z^{-a}/\zeta(a)$ for all $z \in \{1, 2\ldots,\}$, where $\zeta$ is the Riemann Zeta function and $a > 1$ is a control parameter affecting the power-law tail behavior. Second, synthetic data are generated from a random probability measure distributed as the Pitman-Yor prior [41] with a standard Gaussian base distribution and parameters $\lambda > 0$ and $\sigma \in [0, 1)$, as

explained in Appendix A1.5. The parameter $\lambda$ is set to $\lambda = 5000$, while $\sigma$ is varied. For $\sigma = 0$ the Pitman-Yor prior reduces to the Dirichlet prior [42], while $\sigma > 0$ results in heavier tails.

Three benchmark methods are considered. The first one is the *classical* 95% lower bound in Theorem 1. The second one is the *Bayesian* method of [13], which assumes a non-parametric Dirichlet process prior for the distribution of the data stream, estimates its scaling parameter by maximizing the marginal likelihood of the observed sketch, and then computes the posterior of the queried frequencies conditional on the observed sketch. The performance of the lower 5% posterior quantile is compared to the lower bounds obtained with the other methods according to the frequentist metrics defined above. The third benchmark is the bootstrap method of [11], which is nearly exact and optimal for the vanilla CMS (up to some possible finite-sample discrepancy between the bootstrap and population distributions) but is not theoretically valid for other sketching techniques.

Figure 1 compares the performance of the conformal method to those of the three benchmarks on the Zipf data. All methods achieve marginal coverage (4), with the exception of the Bayesian approach which in this case is based on a misspecified prior. The length of the confidence intervals indicates the classical bound is very conservative, while the bootstrap and conformal methods provide relative informative bounds, particularly when $a$ is larger and hash collisions become rarer. The conformal intervals can be the shortest ones, especially if implemented with the adaptive conformity scores. This should not be surprising because the bootstrap may not be optimal for the CMS-CU. Indeed, as shown in Figure 2, the machine learning model deployed by our adaptive scores can take advantage of the fact that this non-linear sketch allows increased precision for more common queries.

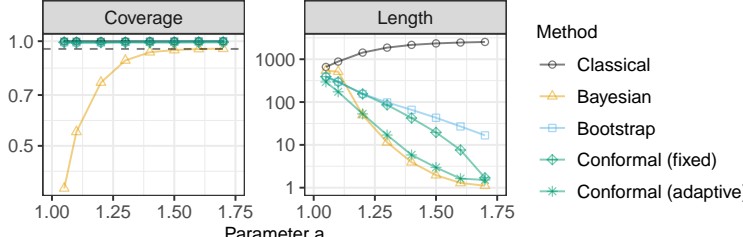

Figure 1: Coverage and length of 95% confidence intervals with data from a Zipf distribution, sketched with CMS-CU. The results are shown as a function of the Zipf tail parameter $a$. Standard errors would be too mall to be clearly visible in this figure, and are hence omitted.

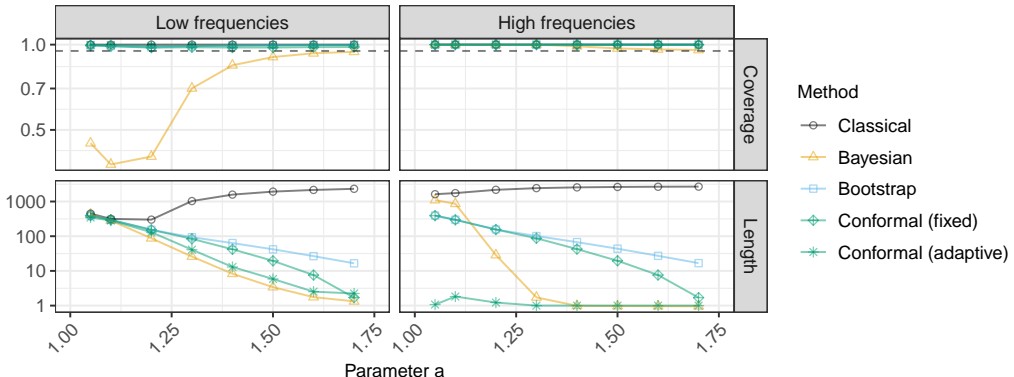

Figure 2: Performance of confidence intervals stratified by the true query frequency. Left: frequency below median; right: frequency above median. Other details are as in Figure 1.

Supplementary results reported in Appendix A3 show the CMS-CU leads to more precise queries with all methods compared to the vanilla CMS; see Figure A1. Figure A2 confirms that conformal lower bounds no longer have a clear advantage over the bootstrap ones if the data are sketched with the vanilla CMS instead of the CMS-CU. In fact, although the conformal intervals obtained with the

adaptive conformity scores can be little shorter than the bootstrap ones even for the vanilla CMS, the latter have the advantage of (approximately, in the limit of large samples) satisfying an even stronger frequency-conditional coverage property equivalent to (8) with bins of size 1 [11]. Analogous results for the experiments with Pitman-Yor process data are also in Appendix A3; see Figures A3–A6.

## 4.2 Analysis of 16-mers in SARS-CoV-2 DNA sequences

This application involves a data set of nucleotide sequences from SARS-CoV-2 viruses made publicly available by the National Center for Biotechnology Information [43]. The data include 43,196 sequences, each consisting of approximately 30,000 nucleotides. The goal is to estimate the empirical frequency of each possible *16-mer*, a distinct sequence of 16 DNA bases in contiguous nucleotides. Given that each nucleotide has one of 4 bases, there are $4^{16} \approx 4.3$ billion possible 16-mers. Thus, exact tracking of all 16-mers is not unfeasible, which allows us to validate the sketch-based queries. Sequences containing missing values are removed during pre-processing, for simplicity.

The experiments are carried out as in Section 4.1 (processing the 16-mers in a random order to ensure their exchangeability), but a larger sample of size 1,000,000 is sketched, and the width $w$ of the hash functions is varied. Figure 3 compares the performances of all methods as a function of the hash width, in terms of marginal coverage and mean confidence interval width. All methods achieve the desired marginal coverage, except for the Bayesian approach when $w$ is large. For small $w$, all methods return intervals of similar width, because the distribution of SARS-CoV-2 16-mers frequencies is quite concentrated with relatively narrow support (Figure A7), which makes it especially difficult to compress the data without much loss. By contrast, the proposed conformal methods yield noticeably shorter confidence intervals if $w$ is large. Figure A8 reports the same results stratified by the frequency of the queried objects, while Figure A9 confirms the advantage of sketching with the CMS-CU as opposed to the vanilla CMS. Table A1 lists 10 common and 10 rare queries along with their corresponding deterministic upper bounds for $w = 50,000$, comparing the lower bounds obtained with each method. Table A2 shows analogous results with $w = 5,000$.

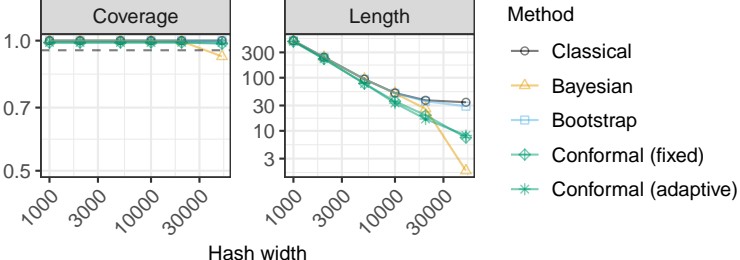

Figure 3: Performance of confidence intervals based on sketched SARS-CoV-2 sequence data. The results are shown as a function of the hash width. Other details are as in Figure 1.

Figure A10 compares the performances of different frequency *point-estimates* in terms of mean absolute deviation from the true frequency. With the classical method, we take the midpoint of the 95% confidence interval as a point estimate, although other approaches are also possible [9]. For the other methods, the point estimate is the lower confidence bound at level $\alpha = 0.5$; in the Bayesian case, this is the posterior median. Although a conformal lower bound at level $\alpha = 0.5$ is not always a reliable estimator of conditional medians [44], because the conformal coverage guarantees treat the query as random, this approach outperformed the benchmarks in all of our experiments.

## 4.3 Analysis of 2-grams in English literature

This application is based on a data set consisting of 18 open-domain classic pieces of English literature downloaded using the NLTK Python package [45] from the Gutenberg Corpus [46]. The goal is to count the frequencies of all *2-grams*—consecutive pairs of English words—across this corpus. After some basic preproccessing to remove punctuation and unusual words (only those in a dictionary of size 25,487 common English words are retained), the total number of 2-grams left in this data set is approximately 1,700,000 (although the total number of all *possible* 2-grams within this dictio-

nary is approximately 650,000,000). The same experiments are then carried out as in Section 4.2, sketching 1,000,000 randomly sampled 2-grams and querying 10,000 independent 2-grams. As in the previous experiments, the 2-grams are processed in a random order to ensure exchangeability.

Figure 4 shows the conformal intervals with adaptive scores achieve the desired coverage and tend to have the shortest width, while the Bayesian intervals are not valid unless the hashes are very wide. The conformal approach has a larger advantage here because these data can be compressed more efficiently compared to the one in the previous section because the frequency distribution of English 2-grams has power-law tails; see Figure A7. Additional results along the lines of those in the previous section are in Appendix A3; see Figures A11–A13 and Tables A1–A2.

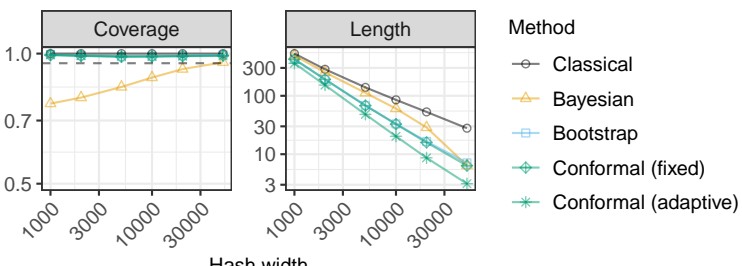

Figure 4: Performance of confidence intervals for random frequency queries, for a sketched data set of English 2-grams in classic English literature. Other details are as in Figure 3.

## 5 Discussion

Conformalized sketching is a non-parametric and data-adaptive statistical method for quantifying uncertainty in problems involving frequency estimation from sketched data. This paper has revolved around the CMS because that is a prominent technique for which several benchmarks are available. However, our method is very broadly applicable because it operates under the sole assumption of data exchangeability, without requiring any knowledge of the sketching algorithm. Of course, exchangeability is not always an appropriate assumption, and thus one may sometimes need to rely on more conservative bounds based on hashing randomness. Yet there are situations in which the sketched data can be seen as exchangeable random samples from a population. For example, in natural language processing one may wish to count the frequencies of non-contiguous tuples of words co-occurring in the same sentence within a large corpus, expanding on the example of Section 4.3. In that case, it would be unfeasible to keep track of all frequencies exactly, but our approach would be applicable as long as the tuples are sketched and queried in a random order. One limitation of conformalized sketching is that it does not provide guarantees about the expected proportion of correct *unique* queries. In fact, if the queries are randomly sampled exchangeably with the sketched data points, some of them may be redundant. Even though the expected proportion of incorrect unique queries is sometimes below $\alpha$ (Figures A14–A15), this is not always the case (Figure A16). It may be possible to modify our procedure to achieve this additional guarantee, but we leave this problem for future work. Future research may also study theoretically, in some settings, the length of our conformal confidence intervals, following for example an approach similar to those of [16, 47].

Accompanying software and data are available online at `https://github.com/msesia/conformalized-sketching`. Experiments were carried out in parallel using a computing cluster; each experiment required less than a few hours with a standard CPU and less than 5GB of memory (20 GB of memory are needed for the analysis of the SARS-CoV-2 DNA data).

**Acknowledgements**

M. S. is supported by NSF grant DMS 2210637 and by an Amazon Research Award. S. F. is also affiliated to IMATI-CNR "Enrico Magenes" (Milan, Italy), and received funding from the European Research Council under the European Union's Horizon 2020 research and innovation programme under grant No 817257. The authors are grateful to three anonymous reviewers for helpful comments and an insightful discussion.

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
