# Appendix to:
# Conformal Frequency Estimation with Sketched Data

**Matteo Sesia**
Department of Data Sciences and Operations
University of Southern California
Los Angeles, California, USA
sesia@marshall.usc.edu

**Stefano Favaro**
Department of Economics and Statistics
University of Torino and Collegio Carlo Alberto
Torino, Italy
stefano.favaro@unito.it

## A1 Additional methodological details

### A1.1 The CMS algorithm

---
**Algorithm A1** CMS

---
**Input:** Data set $Z_1, \ldots, Z_m$. Sketch dimensions $d, w$. Hash functions $h_1, \ldots, h_d$. Query $z$.
**Initialize:** $C_{j,k} = 0$ for all $j \in [d], k \in [w]$.
**for** $i = 1, \ldots, m$ **do**
  **for** $j = 1, \ldots, d$ **do**
    **Increment** $C_{j,h_j(Z_i)} \leftarrow C_{j,h_j(Z_i)} + 1$
  **end for**
**end for**
**Compute** $\hat{f}_{\text{up}}^{\text{CMS}}(z) = \min_{j \in [d]}\{C_{j,h_j(z)}\}$.
**Output:** deterministic upper-bound for the frequency of $z$ in the data set: $\hat{f}_{\text{up}}^{\text{CMS}}(z)$.

---

### A1.2 The CMS-CU algorithm

---
**Algorithm A2** CMS-CU

---
**Input:** Data set $Z_1, \ldots, Z_m$. Sketch dimensions $d, w$. Hash functions $h_1, \ldots, h_d$. Query $z$.
**Initialize:** $C_{j,k} = 0$ for all $j \in [d], k \in [w]$.
**for** $i = 1, \ldots, m$ **do**
  **Compute** $j^* = \arg\min_{j \in [d]} C_{j,h_j(Z_i)}$.
  **Increment** $C_{j^*,h_{j^*}(Z_i)} \leftarrow C_{j^*,h_{j^*}(Z_i)} + 1$
**end for**
**Compute** $\hat{f}_{\text{up}}^{\text{CMS-CU}}(z) = \min_{j \in [d]}\{C_{j,h_j(z)}\}$.
**Output:** deterministic upper-bound for the frequency of $z$ in the data set: $\hat{f}_{\text{up}}^{\text{CMS-CU}}(z)$.

---

### A1.3 Conformalized sketching

---

**Algorithm A3** Conformalized sketching

---

**Input:** Data set $Z_1, \ldots, Z_m$. Sketching function $\phi$. Warm-up duration $m_0 \ll m$.
   A (trainable) rule for computing nested intervals $[\hat{L}_{m,\alpha}(\cdot; t), \hat{U}_{m,\alpha}(\cdot; t)], t \in \mathcal{T}$.
   Number of data points $m_0^{\text{train}} < m_0$ used for training $[\hat{L}_{m,\alpha}(\cdot; t), \hat{U}_{m,\alpha}(\cdot; t)]$.
   A partition $\mathcal{B} = (B_1, \ldots, B_L)$ of $\{0, \ldots, m\}$ into $L$ intervals.
   Random query $Z_{m+1}$. Desired coverage level $1 - \alpha \in (0, 1)$.
**Initialize** a sparse dictionary $f_{m_0}^{\text{wu}}(z) = 0, \forall z \in \mathcal{Z}$.
**for** $i = 1, \ldots, m_0$ **do**
   **Increment** $f_{m_0}^{\text{wu}}(Z_i) \leftarrow f_{m_0}^{\text{wu}}(Z_i) + 1$.
**end for**
**Initialize** a sparse dictionary $f_{m-m_0}^{\text{sv}}(z) = 0, \forall z \in \mathcal{Z}$.
**Initialize** an empty sketch $\phi(\emptyset)$.
**for** $i = m_0 + 1, \ldots, m$ **do**
   **Update** the sketch $\phi$ with the new observation $Z_i$.
   **if** $f_{m_0}^{\text{wu}}(Z_i) > 0$ **then**
      **Increment** $f_{m-m_0}^{\text{sv}}(Z_i) \leftarrow f_{m-m_0}^{\text{sv}} + 1$.
   **end if**
**end for**
**Train** $[\hat{L}_{m,\alpha}(\cdot; t), \hat{U}_{m,\alpha}(\cdot; t)]$ using the data in $\{(X_i, Y_i)\}_{i=1}^{m_0^{\text{train}}}$.
**for** $i = m_0^{\text{train}} + 1, \ldots, m_0$ **do**
   **Set** $X_i = (Z_i, \phi(Z_{m_0+1}, \ldots, Z_m))$ as in (12).
   **Set** $Y_i = f_{m-m_0}^{\text{sv}}(Z_i)$.
   **Compute** the conformity score $E(X_i, Y_i)$ with (5), using $[\hat{L}_{m,\alpha}(\cdot; t), \hat{U}_{m,\alpha}(\cdot; t)]$.
   **Assign** each score $E(X_i, Y_i)$ to an appropriate frequency bin $B \in \mathcal{B}$ based on $Y_i$.
**end for**
**for** $l = 1, \ldots, L$ **do**
   **Compute** $\hat{Q}_{n_l, 1-\alpha}(B_l)$ as the $\lceil (1 - \alpha)(n_l + 1) \rceil$ smallest value among the $n_l$ scores in bin $B_l$.
**end for**
**Set** $\hat{Q}_{n, 1-\alpha}^* = \max_l \hat{Q}_{n_l, 1-\alpha}(B_l)$.
**Set** $X_{m+1} = (Z_{m+1}, \phi(Z_{m_0+1}, \ldots, Z_m))$ as in (12).
**Output:** a $(1 - \alpha)$-level confidence interval

$$\left[ f_{m_0}^{\text{wu}}(Z_{m+1}) + \hat{L}_{m,\alpha}(X_{m+1}; \hat{Q}_{n, 1-\alpha}^*), f_{m_0}^{\text{wu}}(Z_{m+1}) + \hat{U}_{m,\alpha}(X_{m+1}; \hat{Q}_{n, 1-\alpha}^*) \right]$$

for the unobserved frequency $f_m(Z_{m+1})$ of $Z_{m+1}$ defined in (2).

---

### A1.4 Constructing two-sided conformal confidence intervals

This section describes two alternatives methods for constructing two-sided conformal confidence intervals. The first method, explained in Section A1.4.1, consists of directly calibrating a sequence of nested two-sided intervals, as outlined in Section 3.3. The second method, explained in Section A1.4.2, consists of separately calibrating two sequences of lower and upper one-sided confidence intervals, each adopting the significance level $\alpha/2$ instead of $\alpha$. The second approach is easier to implement compared to the first one, building upon the techniques detailed earlier in this paper, but it may be less statistically efficient.

### A1.4.1 Construction based on conditional histograms

Two-sided conformal confidence intervals for $f_m(X_{m+1})$ can be constructed by following the general recipe outlined in Section 3.3. To implement this method practically, one needs to fix an increasing sequence of candidate intervals $[\hat{L}_{m,\alpha}(\cdot; t), \hat{U}_{m,\alpha}(\cdot; t)]$, depending on $Z_{m+1}$ and $\phi(Z_{m_0+1}, \ldots, Z_m)$. Possible choices for such sequence may be directly borrowed from the existing literature on conformal inference for regression, including for example the quantile regression approach of [3] or the conditional histogram approach of [4]. Here, we describe a particular imple-

mentation that combines the idea in [4] with a Bayesian model, in continuity with the works of [1, 2] on Bayesian empirical frequency estimation from sketched data. However, the same idea could easily accommodate a quantile regression model or any other machine learning algorithm instead of the Bayesian model, as explained in [4]. Note that the following paragraphs largely retrace the same steps as in [4], which are however useful to recap here to make the presentation self contained.

For any $j \in [m]$, let $\hat{\varphi}_j(x)$ indicate the posterior probability of $f_m(X_{m+1}) = j$ for $X_{m+1} = x$ as estimated by any Bayesian model for frequency estimation given sketched data, such as that of [1] based on a Dirichlet process prior, for example. For convenience of notation, we will sometimes refer to the full posterior distribution of $f_m(X_{m+1})$ simply as $\hat{\varphi}$. Note that, in general, the form of the posterior distribution $\hat{\varphi}$ may depend on $m$ as well as on the sketched data in $\phi(Z_{m_0+1}, \ldots, Z_m)$. Following in the footsteps of [4], define the following bi-valued function $\mathcal{S}$ taking as input a query $x$, the posterior distribution $\hat{\varphi}$, a scalar threshold $t \in [0, 1]$, and two intervals $S^-, S^+ \subseteq \{1, \ldots, m\}$:

$$\mathcal{S}(x, \hat{\varphi}, S^-, S^+, t) := \underset{(l,u) \in \{1, \ldots, m\}^2 \,:\, l \leq u}{\arg \min} \left\{ |u - l| : \sum_{j=l}^{u} \hat{\varphi}_j(x) \geq t, \; S^- \subseteq [l, u] \subseteq S^+ \right\}. \quad (1)$$

Above, it is implied that we choose the value of $(l, u)$ minimizing $\sum_{j=l}^{u} \hat{\varphi}_j(x)$ among the feasible ones with minimal $|u - l|$, whenever the optimization problem does not have a unique solution. Therefore, we can assume without loss of generality that (1) has a unique solution; if that is not the case, we can break the ties at random by adding a little noise to $\hat{\varphi}$. As explained in [4], the problem defined in (1) can be solved efficiently, at computational cost linear in $m$. Note that we will sometimes refer to sub-intervals of $[m]$ as either contiguous subsets of $\{1, \ldots, m\}$ (e.g., $S^-$) or as pairs of lower and upper endpoints (e.g., $[l, u]$).

If $S^- = \emptyset$ and $S^+ = \{1, \ldots, m\}$, the expression in (1) computes the shortest possible interval with total posterior probability mass above $t$. In general, the optimization in (1) involves the additional *nesting* constraint that the output $\mathcal{S}$ must satisfy $S^- \subseteq \mathcal{S} \subseteq S^+$, which will be needed to guarantee the resulting sequence of confidence intervals indexed by $t$ is nested. Note that the inequality in (1) involving $t$ may not be binding at the optimal solution due to the discrete nature of the optimization problem. However, the above construction could be easily modified by introducing some suitable randomization leading to confidence intervals that are even tighter on average, as explained in [4].

For any integer $T \geq 1$, consider an increasing sequence $t_\tau \in [0, 1]$, for $\tau \in \{0, \ldots, T\}$. A nested sequence of $T$ intervals indexed by $\tau \in \{0, \ldots, T\}$, which may be written in the form of

$$S_t = \left[ \hat{L}_{m,\alpha}(X_{m+1}; t_\tau), \hat{U}_{m,\alpha}(X_{m+1}; t_\tau) \right],$$

for appropriate lower and upper endpoints $\hat{L}_{m,\alpha}(X_{m+1}; t_\tau)$ and $\hat{U}_{m,\alpha}(X_{m+1}; t_\tau)$, respectively, is then constructed from (1) as follows. First, fix any *starting index* $\bar{\tau} \in \{0, 1, \ldots, T\}$ and define $S_{\bar{\tau}}$ by applying (1) without the nesting constraints (with $S^- = \emptyset$ and $S^+ = \{1, \ldots, m\}$):

$$S_{\bar{\tau}} := \mathcal{S}(x, \hat{\varphi}, \emptyset, \{1, \ldots, m\}, t_{\bar{\tau}}), \quad (2)$$

Note the explicit dependence on $x$ and $\hat{\varphi}$ of the left-hand-side above is omitted for simplicity, although it is important to keep in mind that $S_{\bar{\tau}}$ does of course depend on these quantities.

Having computed the initial interval $S_{\bar{\tau}}$, we recursively extend the definition to the wider intervals indexed by $\tau = \bar{\tau} + 1, \ldots, T$ as follows:

$$S_\tau := \mathcal{S}(x, \hat{\varphi}, S_{\tau-1}, \{1, \ldots, m\}, t_\tau).$$

See [4] for a schematic visualization of this step. Similarly, the narrower intervals $S_\tau$ indexed by $\tau = \bar{\tau} - 1, \bar{\tau} - 2, \ldots, 0$ are defined recursively as:

$$S_\tau := \mathcal{S}(x, \hat{\varphi}, \emptyset, S_{\tau+1}, t_\tau).$$

See [4] for a schematic visualization of this step. As a result of this construction, the sequence of intervals $\{S_\tau\}_{\tau=0}^{T}$ is nested regardless of the starting point $\bar{\tau}$ in (2), for which a typical choice is such that $t_{\bar{\tau}} = 1 - \alpha$. Then, two-sided conformal confidence intervals for $f_m(X_{m+1})$ can be obtained by applying Algorithm A3 with this particular sequence of input nested intervals. We refer to [4] for further details on the construction of nested intervals outlined above.

### A1.4.2 Construction based on a pair of one-sided intervals with Bonferroni correction

An alternative, and somewhat simpler, approach to building two-sided conformal confidence intervals for $f_m(X_{m+1})$ at level $1 - \alpha$ consists of constructing a pair of lower and upper one-sided confidence intervals at level $1 - \alpha/2$. In particular, consider the following two nested sequences $S_t^l$ and $S_t^u$ of one-sided confidence intervals, each indexed by a scalar parameter $t$:

$$S_t^l = [\hat{L}_{m,\alpha/2}(X_{m+1}; t), \hat{f}_{\text{up}}^{\text{CMS}}(X_{m+1})], \qquad S_t^u = [0, \hat{U}_{m,\alpha/2}(X_{m+1}; t)],$$

where $\hat{f}_{\text{up}}^{\text{CMS}}(X_{m+1})$ is a deterministic upper bound for the unknown true empirical frequency of $X_{m+1}$; e.g., see Section 1.2. The sequences $S_t^l$ and $S_t^u$ can be separately calibrated using the conformal inference method described in Sections 3.3 and 3.4, for any given choice of frequency-range partition $\mathcal{B}$, as we shall make more precise below. This gives two distinct data-adaptive thresholds $\hat{Q}_{n,1-\alpha/2}^{*,l}$ and $\hat{Q}_{n,1-\alpha/2}^{*,u}$, respectively, such that, $\forall B \in \mathcal{B}$,

$$\mathbb{P}\left[ f_m(X_{m+1}) \geq \hat{L}_{m,\alpha/2}(X_{m+1}; \hat{Q}_{n,1-\alpha/2}^{*,l}) \mid f_m(Z_{m+1}) \in B \right] \geq 1 - \frac{\alpha}{2},$$

and

$$\mathbb{P}\left[ f_m(X_{m+1}) \leq \hat{U}_{m,\alpha/2}(X_{m+1}; \hat{Q}_{n,1-\alpha/2}^{*,u}) \mid f_m(Z_{m+1}) \in B \right] \geq 1 - \frac{\alpha}{2}.$$

By a union bound, we obtain that the following two-sided conformal confidence interval has valid coverage, in the sense of (8), at level $1 - \alpha$:

$$[\hat{L}_{m,\alpha/2}(X_{m+1}; \hat{Q}_{n,1-\alpha/2}^{*,l}), \hat{U}_{m,\alpha/2}(X_{m+1}; \hat{Q}_{n,1-\alpha/2}^{*,u})].$$

Different practical implementations are available to construct the sequences of candidate lower bounds $\hat{L}_{m,\alpha/2}(X_{m+1}; t)$ and upper bounds $\hat{U}_{m,\alpha/2}(X_{m+1}; t)$. Two concrete examples are explained below.

**Constant conformity scores.** A simple option to construct the sequence $\hat{L}_{m,\alpha/2}(X_{m+1}; t)$ is to directly apply the method described in Section 3.4, for example by shifting $\hat{f}_{\text{up}}^{\text{CMS}}(X_{m+1})$ downward by a constant $t$. Then, the conformalized threshold $\hat{Q}_{n,1-\alpha/2}^{*,l}$ can be calibrated exactly as described in Section 3.3. The sequence of candidate upper bounds $\hat{U}_{m,\alpha/2}(X_{m+1}; t)$ can also be constructed similarly to $\hat{L}_{m,\alpha/2}(X_{m+1}; t)$, for example by adding a constant $t$ to the trivial lower bound of 0, up to the deterministic upper bound $\hat{f}_{\text{up}}^{\text{CMS}}(X_{m+1})$. The threshold $\hat{Q}_{n,1-\alpha/2}^{*,u}$ for $\hat{U}_{m,\alpha/2}(X_{m+1}; t)$ can then be calibrated as usual with Algorithm A3.

**Bootstrap conformity scores.** An alternative option to construct the sequence $\hat{L}_{m,\alpha/2}(X_{m+1}; t)$ consists of shifting downward by a constant $t$ the bootstrap lower bound calculated with the method of [5], at level $\alpha/2$. Similarly, the sequence $\hat{U}_{m,\alpha/2}(X_{m+1}; t)$ can be obtained by shifting upward by a constant $t$ the analogous bootstrap upper bound at level $1 - \alpha/2$. Thus, in the special case of the vanilla CMS, our conformal confidence intervals based on these scores intuitively become very similar to the bootstrap confidence intervals of [5]. In general, however, the difference remains that the intervals of [5] rely on the linearity of the CMS, while ours are theoretically valid regardless of how the data are sketched. We have observed this option works well in practice, at least within the scope of our numerical experiments. Therefore, this is the implementation adopted in our numerical experiments described in Section A4.

### A1.5 Sampling from a Pitman-Yor predictive distribution

The data points are sampled sequentially from the following predictive distribution, which has parameters $\lambda > 0$ and $\sigma \in [0, 1)$. After sampling $Z_1$ from a standard normal distribution, $\mathcal{N}(0, 1)$, fix any $i \geq 1$ and let $Z_1, \ldots, Z_i$ indicate the data stream observed up to that point. Denote by $k_i$ the number of distinct elements within it, and by $V_i = (V_{i,1}, \ldots, V_{i,k_i})$ the set of such distinct values. Further, let $c_{i,l}$ indicate the number of times that object $V_{i,l}$ has been observed in $Z_1, \ldots, Z_i$, for $l \in \{1, \ldots, k_i\}$. Then, $Z_{i+1}$ is generated as follows:

$$Z_{i+1} \mid Z_1, \ldots, Z_i = \begin{cases} V_{i,l}, & \text{with probability } \frac{c_{i,l} - \sigma}{\lambda + i}, \text{ for } l \in \{1, \ldots, k_i\}, \\ \mathcal{N}(0, 1), & \text{with probability } \frac{\lambda + k_i \sigma}{\lambda + i}. \end{cases}$$

Above, the second case which occurs with probability $(\lambda + k_i\sigma)/(\lambda + i)$ corresponds to sampling a new unique value from the standard normal distribution.

## A2 Mathematical proofs

### A2.1 Proof of Proposition 1

*Proof.* Consider $((X_{\pi(1)}, Y_{\pi(1)}), \ldots, (X_{\pi(m_0)}, Y_{\pi(m_0)}), (X_{\pi(m+1)}, Y_{\pi(m+1)}))$ for any permutation $\pi$ of $\{1, \ldots, m_0, m+1\}$. This is equal to $((X_1', Y_1'), \ldots, (X_{m_0}', Y_{m_0}'), (X_{m+1}', Y_{m+1}'))$, defined by applying the functions in (11)–(12) to a shuffled data set $Z_{\tilde{\pi}(1)}, \ldots, Z_{\tilde{\pi}(m+1)}$, where $\tilde{\pi}$ indicates a permutation of $\{1, \ldots, m+1\}$ that agrees with $\pi$ on $\{1, \ldots, m_0, m+1\}$ and leaves $\{m_0+1, \ldots, m\}$ unchanged. Therefore,

$$
\begin{aligned}
((X_{\pi(1)}&, Y_{\pi(1)}), \ldots, (X_{\pi(m_0)}, Y_{\pi(m_0)}), (X_{\pi(m+1)}, Y_{\pi(m+1)})) \\
&= ((X_1', Y_1'), \ldots, (X_{m_0}', Y_{m_0}'), (X_{m+1}', Y_{m+1}')) \\
&\stackrel{d}{=} ((X_1, Y_1), \ldots, (X_{m_0}, Y_{m_0}), (X_{m+1}, Y_{m+1})),
\end{aligned}
$$

where the last equality in distribution follows directly from the assumption that $Z_1, \ldots, Z_{m+1}$ are exchangeable. $\qquad\square$

### A2.2 Proof of Theorem 2

*Proof.* The following notation will be helpful: let $B(Y_i) \in \mathcal{B}$ indicate the frequency bin into which $Y_i$ belongs, for $i \in \{1, \ldots, m_0, m+1\}$. We begin by proving the result for the simpler case in which Algorithm A3 is applied using conformity scores that do not require training, in which case $m_0^{\text{train}} = 0$. For $i \in \{1, \ldots, m_0, m+1\}$, define the random variables $Y_i$ and $X_i$ as in (11)–(12), respectively. We already know from Proposition 1 that $(X_1, Y_1), \ldots, (X_{m_0}, Y_{m_0}), (X_{m+1}, Y_{m+1})$ are exchangeable. This implies that the conformity scores $E(X_i, Y_i)$ are exchangeable with one another, for $i \in \{1, \ldots, m_0, m+1\}$, because each of them only depends on $X_i, Y_i$ and on the separate data points in the sketch $\phi(Z_{m_0+1}, \ldots, Z_m)$. Therefore, $E_{m+1}$ is also exchangeable with the subset of conformity scores with indices in $\{i \in \{1, \ldots, m_0\} : B(Y_i) = B(Y_{m+1})\}$. Now, fix any bin $B^* \in \mathcal{B}$ and assume $B(Y_{m+1}) = B^*$. Now, note that the interval output by Algorithm A3 does not cover the true frequency $f_m(Z_{m+1})$ if and only if $E_{m+1} > \hat{Q}_{n,1-\alpha} \geq \hat{Q}_{n_l,1-\alpha}(B^*)$. However, a standard exchangeability argument for the conformity scores in $\{i \in \{1, \ldots, m_0\} : B(Y_i) = B^*\}$ shows that $\mathbb{P}[E_{m+1} > \hat{Q}_{n_l,1-\alpha}(B^*) \mid B(Y_{m+1}) = B^*] \leq 1 - \alpha$; for example, see Lemma 1 of [3]. This completes the first part of the proof. The second part with $m_0^{\text{train}} > 0$ follows very similarly: Proposition 1 implies that $(X_{m_0^{\text{train}}+1}, Y_{m_0^{\text{train}}+1}), \ldots, (X_{m_0}, Y_{m_0}), (X_{m+1}, Y_{m+1})$ are exchangeable, and so must be the conformity scores $E_i$ for $i \in \{m_0^{\text{train}} + 1, \ldots, m_0, m+1\}$ because each of them only depends on the corresponding $X_i, Y_i$ and on the separate set of observations indexed by $\{1, \ldots, m_0^{\text{train}}\}$, as well as on the sketch $\phi(Z_{m_0+1}, \ldots, Z_m)$. The rest of the proof is exactly the same as in the first part because the empirical quantiles $\hat{Q}_{n_l,1-\alpha}(B)$ are only computed on subsets of the data indexed by $\{m_0^{\text{train}} + 1, \ldots, m_0\}$. $\qquad\square$

## A3 Supplementary figures and tables

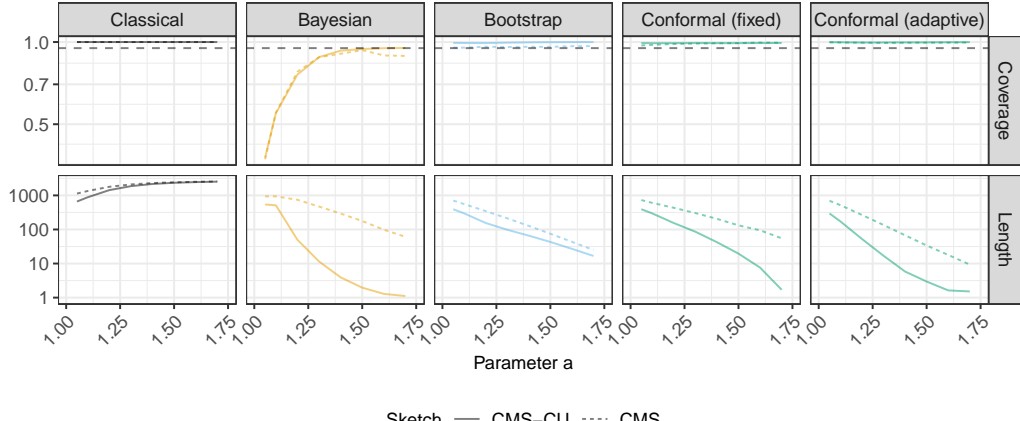

Figure A1: Performance of 95% confidence intervals for random frequency queries, based on synthetic data from a Zipf distribution. The data are sketched with either the vanilla CMS or the CMS-CU. The results are shown as a function of the Zipf tail parameter $a$. Other details are as in Figure 1.

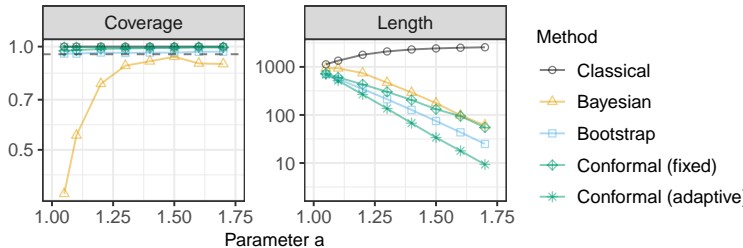

Figure A2: Performance of 95% confidence intervals for random frequency queries, based on synthetic data from a Zipf distribution, sketched with the vanilla CMS. The results are shown as a function of the Zipf tail parameter $a$. Other details are as in Figure 1.

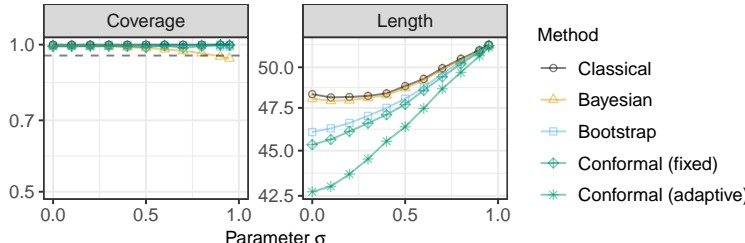

Figure A3: Empirical coverage and length of 95% confidence intervals for random frequency queries on a synthetic data set sampled from the predictive distribution of a Pitman-Yor process. The data are sketched with the CMS-CU. The results are shown as a function of the Pitman-Yor process parameter $\sigma$. Other details are as in Figure 1.

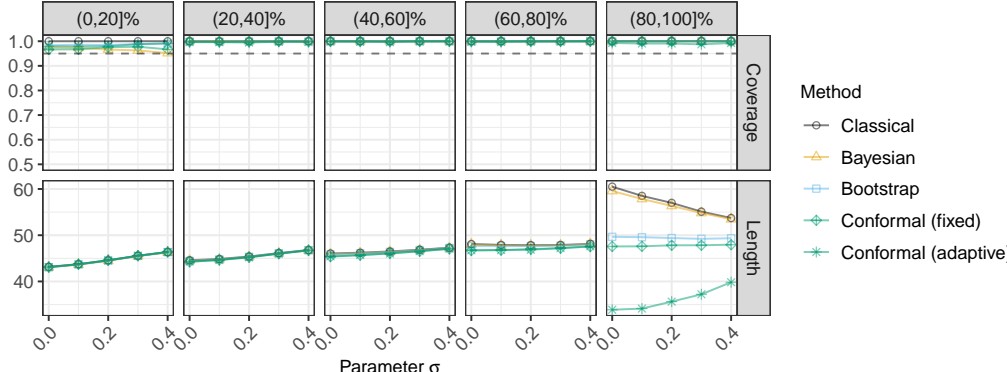

Figure A4: Performance of 95% confidence intervals for random frequency queries on a synthetic data set sampled from the predictive distribution of a Pitman-Yor process. The results are stratified by the quintile of the true query frequency. Other details are as in Figure A3.

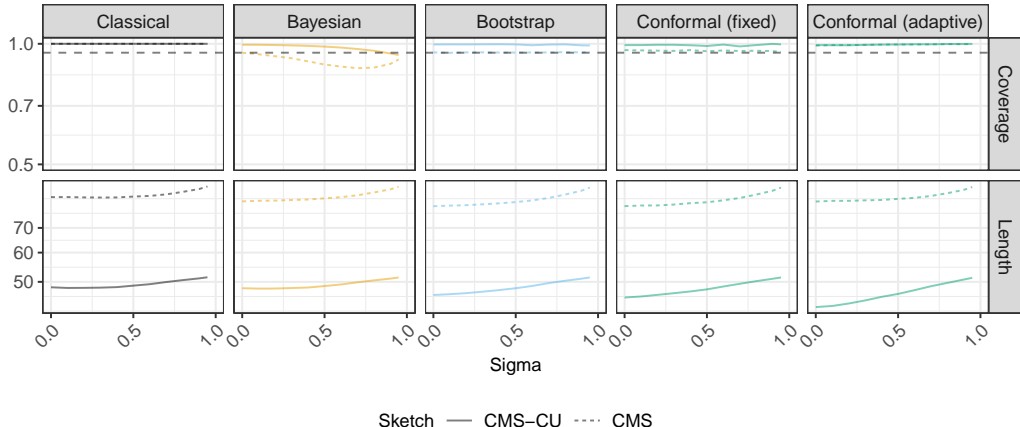

Figure A5: Performance of 95% confidence intervals for random frequency queries, based on synthetic data sampled from the predictive distribution of a Pitman-Yor process and sketched with either the vanilla CMS or the CMS-CU. The results are shown as a function of the Pitman-Yor process parameter $\sigma$. Other details are as in Figure A3.

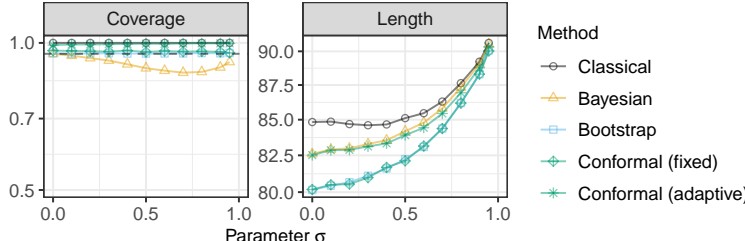

Figure A6: Performance of 95% confidence intervals for random frequency queries, based on synthetic data sampled from the predictive distribution of a Pitman-Yor process and sketched with the vanilla CMS. The results are shown as a function of the Pitman-Yor process parameter $\sigma$. Other details are as in Figure A3.

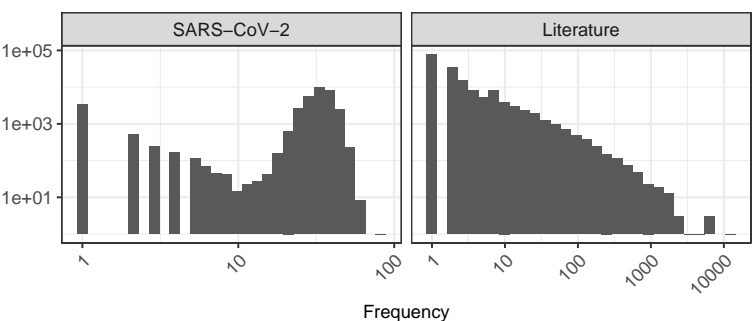

Figure A7: True frequency distribution of unique objects in two real data sets. Left: sequenced SARS-CoV-2 DNA 16-mers. Right: English 2-grams in a corpus of classic English literature.

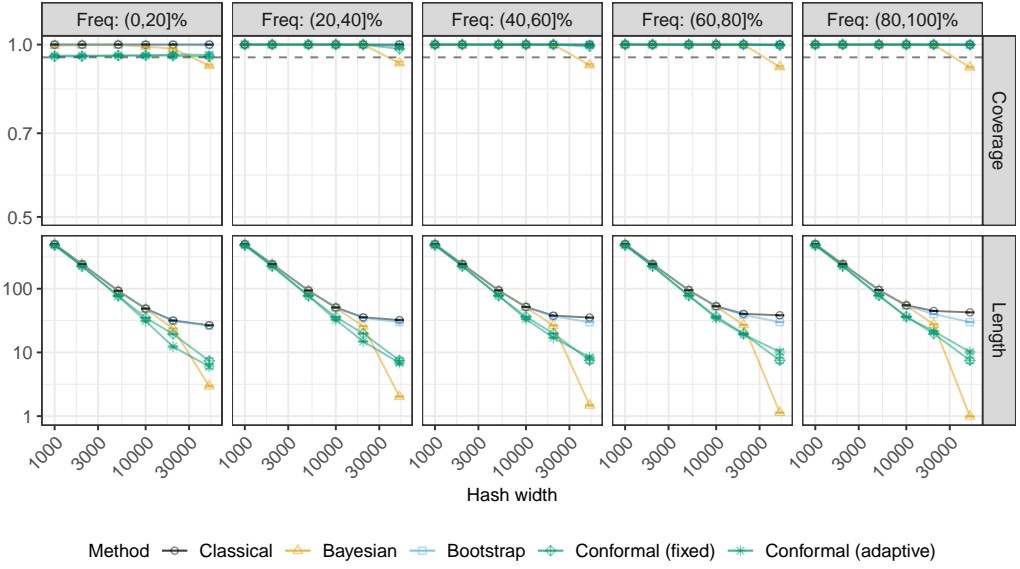

Figure A8: Performance of 95% confidence intervals for random frequency queries on SARS-CoV-2 sequence data sketched with the CMS-CU. The results are shown as a function of the hash width and stratified by the quintile of the true query frequency. Other details are as in Figure 3.

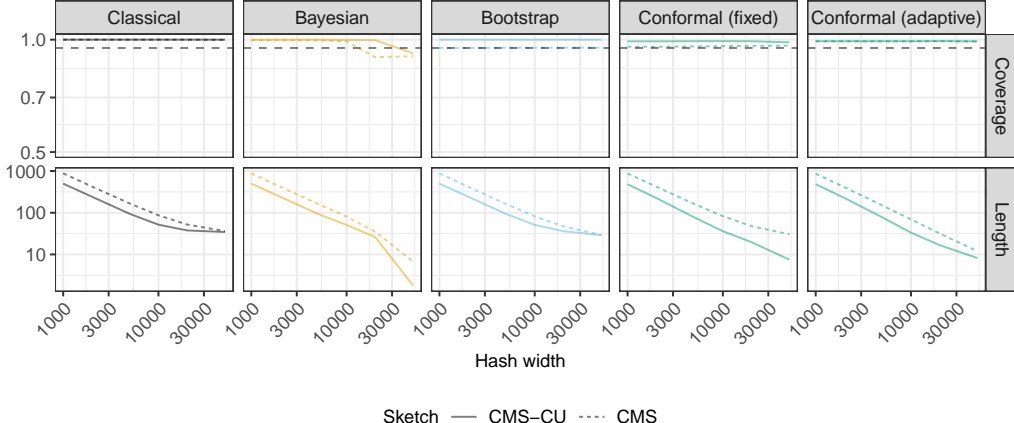

Figure A9: Performance of 95% confidence intervals for random frequency queries on SARS-CoV-2 sequence data. The data are sketched with either the vanilla CMS or the CMS-CU. The results are shown as a function of the hash width. Other details are as in Figure 3.

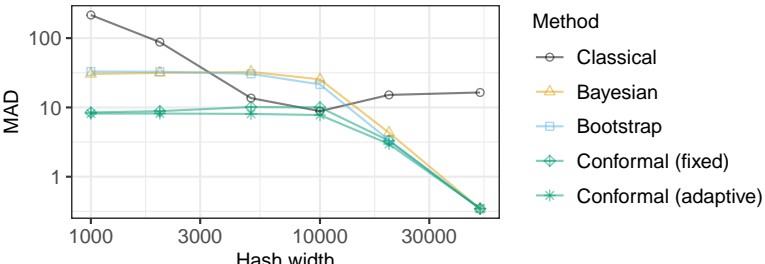

Figure A10: Median absolute deviation of point estimates for random frequency queries on SARS-CoV-2 sequence data sketched with the CMS-CU. The results are shown as a function of the hash width. Other details are as in Figure 3.

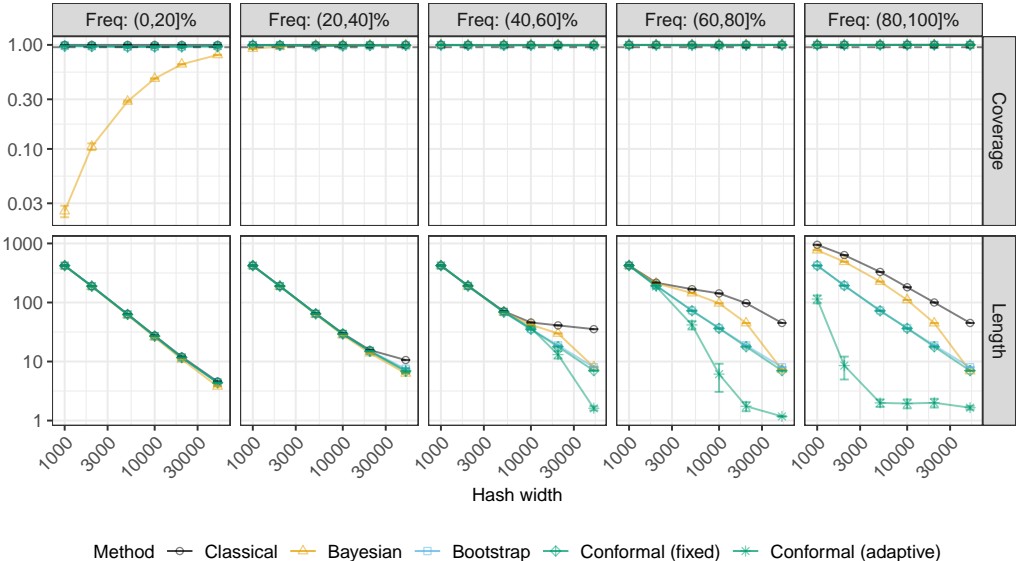

Figure A11: Performance of 95% confidence intervals for random frequency queries on a data set of 2-grams in classic English literature, sketched with the CMS-CU. The results are are shown as a function of the hash width and stratified by the quintile of the true query frequency. Other details are as in Figure 4.

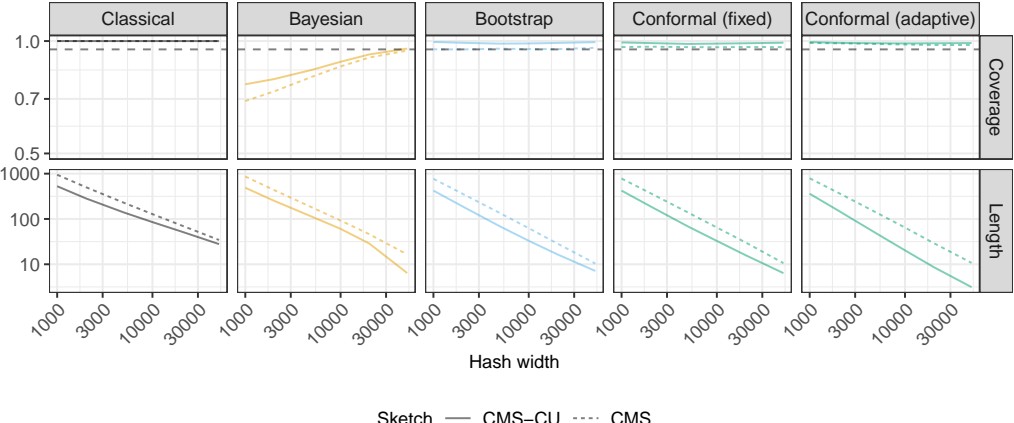

Figure A12: Performance of 95% confidence intervals for random frequency queries on a data set of 2-grams in classic English literature. The data are sketched with either the vanilla CMS or the CMS-CU. The results are shown as a function of the hash width. Other details are as in Figure 4.

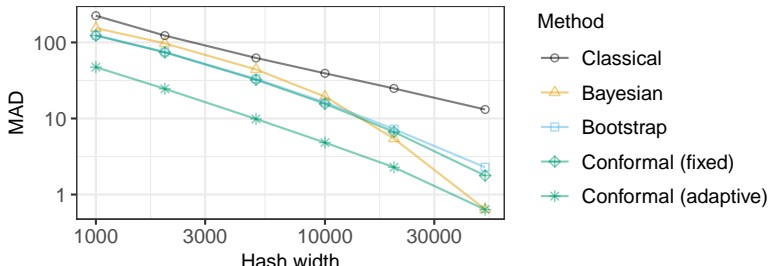

Figure A13: Median absolute deviation of point estimates for random frequency queries on a data set of 2-grams in classic English literature, sketched with the CMS-CU. The results are shown as a function of the hash width. Other details are as in Figure 4.

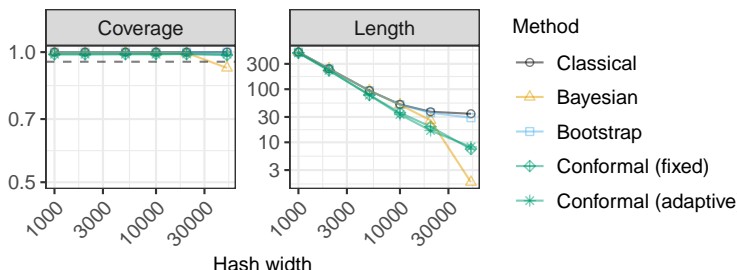

Figure A14: Performance of 95% confidence intervals for random frequency queries, on a sketched data set of 2-grams in classic English literature, keeping only unique queries. The coverage is defined as the empirical proportion of unique queries whose frequency is correctly covered by the output confidence intervals. The data are sketched with the CMS-CU. The results are shown as a function of the hash width. Other details are as in Figure 3.

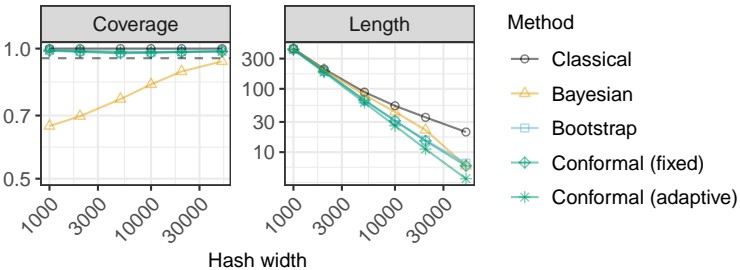

Figure A15: Performance of 95% confidence intervals for random frequency queries, on a sketched data set of 2-grams in classic English literature, keeping only unique queries. The coverage is defined as the empirical proportion of unique queries whose frequency is correctly covered by the output confidence intervals. The results are shown as a function of the hash width. Other details are as in Figure 4.

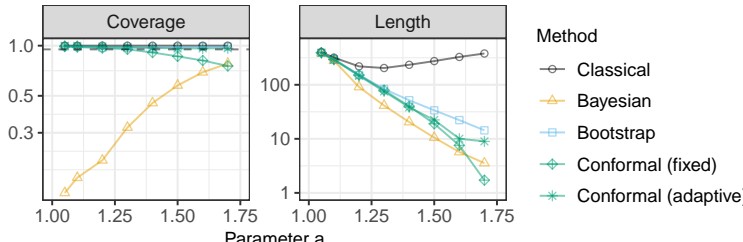

Figure A16: Performance of confidence intervals for random frequency queries, keeping only unique queries. The coverage is defined as the empirical proportion of unique queries whose frequency is correctly covered by the output confidence intervals. The results are shown as a function of the Zipf tail parameter $a$. Other details are as in Figure 1.

Table A1: True frequencies, deterministic upper bounds, and 95% lower bounds for some a few random queries in two sketched data sets. Sketching with CMS-CU with $w = 50,000$. Lower bounds written in green are below the true frequency; those in red are above. For each query, the highest lowest bound below the true frequency is highlighted in bold.

| | | | 95% Lower bound | | | | |
| | | | | | | Conformal | |
| Data | Frequency | Upper bound | Classical | Bayesian | Bootstrap | Fixed | Adaptive |
|---|---|---|---|---|---|---|---|
| **SARS-CoV-2** | | | | | | | |
| AATTATTATAAGAAAG | 81 | 81 | 26 | **81** | 52 | 50 | 36 |
| TCAGACAACTACTATT | 76 | 76 | 21 | **55** | 47 | 45 | 32 |
| AAAGTTGATGGTGTTG | 73 | 73 | 18 | **59** | 44 | 42 | 31 |
| CAATTATTATAAGAAA | 63 | 63 | 8 | **48** | 34 | 32 | 26 |
| ATCAGACAACTACTAT | 60 | 60 | 5 | **44** | 31 | 29 | 26 |
| ACCTTTGACAATCTTA | 55 | 55 | 0 | **52** | 26 | 24 | 27 |
| ATTTGAAGTCACCTAA | 55 | 55 | 0 | **55** | 26 | 24 | 27 |
| CATGCAAATTACATAT | 54 | 54 | 0 | **54** | 25 | 23 | 26 |
| GAATTTCACAGTATTC | 54 | 54 | 0 | **54** | 25 | 23 | 27 |
| TTTGTAGAAAACCCAG | 53 | 53 | 0 | **53** | 24 | 22 | 27 |
| AGTTGCAGAGTGGTTT | 24 | 24 | 0 | 13 | 0 | 0 | **20** |
| TCTTCACAATTGGAAC | 24 | 24 | 0 | 12 | 0 | 1 | **20** |
| TTCTGCTCGCATAGTG | 24 | 24 | 0 | 12 | 0 | 0 | **20** |
| CTACTTTAGATTCGAA | 23 | 23 | 0 | 11 | 0 | 0 | **19** |
| GCTGGTGTCTCTATCT | 23 | 23 | 0 | **23** | 0 | 1 | 19 |
| TTCTAAGAAGCCTCGG | 23 | 24 | 0 | 14 | 0 | 0 | **20** |
| GGGCTGTTGTTCTTGT | 22 | 24 | 0 | 12 | 0 | 0 | **20** |
| ACGTTCGTGTTGTTTT | 20 | 20 | 0 | **20** | 0 | 0 | 16 |
| GAAGTCTTTGAATGTG | 20 | 20 | 0 | **20** | 0 | 0 | 16 |
| CAAACCTGGTAATTTT | 3 | 3 | 0 | **3** | 0 | 0 | 0 |
| **Literature** | | | | | | | |
| of the | 12565 | 12568 | 12513 | 12544 | 12557 | 12556 | **12562** |
| in the | 6188 | 6190 | 6135 | 6169 | 6179 | 6179 | **6180** |
| and the | 6173 | 6175 | 6120 | 6151 | 6164 | 6164 | **6165** |
| the of | 6015 | 6017 | 5962 | 5990 | 6006 | 6006 | **6007** |
| the lord | 4186 | 4195 | 4140 | 4165 | 4184 | 4184 | 4184 |
| to the | 3465 | 3467 | 3412 | 3445 | 3456 | 3456 | **3463** |
| the and | 2250 | 2251 | 2196 | 2227 | 2240 | 2240 | **2248** |
| all the | 2226 | 2230 | 2175 | 2207 | 2219 | 2219 | **2224** |
| and he | 2169 | 2173 | 2118 | 2153 | 2162 | 2162 | **2167** |
| to be | 2062 | 2064 | 2009 | 2043 | 2053 | 2053 | **2060** |
| man on | 22 | 29 | 0 | 10 | 18 | 18 | 18 |
| their hand | 22 | 24 | 0 | 9 | 13 | 13 | 0 |
| no need | 20 | 28 | 0 | 9 | 17 | 17 | 16 |
| and brother | 12 | 14 | 0 | 2 | 3 | 3 | 0 |
| miss would | 10 | 13 | 0 | **3** | 2 | 2 | 0 |
| i please | 8 | 12 | 0 | **3** | 1 | 1 | 1 |
| also how | 3 | 13 | 0 | 2 | 2 | 2 | 0 |
| in under | 3 | 9 | 0 | **2** | 0 | 0 | 0 |
| ten old | 3 | 6 | 0 | **1** | 0 | 0 | 0 |
| fault he | 1 | 9 | 0 | **1** | 0 | 0 | 0 |

Table A2: True frequencies, upper and lower bounds for a few random queries in two sketched data sets. Hash width $w = 50,000$. Other details are as in Table A1.

| Data | Frequency | Upper bound | 95% Lower bound | | | Conformal | |
| --- | --- | --- | --- | --- | --- | --- | --- |
| | | | Classical | Bayesian | Bootstrap | Fixed | Adaptive |
| **SARS-CoV-2** | | | | | | | |
| AATTATTATAAGAAAG | 81 | 209 | 0 | 4 | 0 | 0 | 18 |
| TCAGACAACTACTATT | 76 | 213 | 0 | 8 | 0 | 0 | 18 |
| AAAGTTGATGGTGTTG | 73 | 130 | 0 | 2 | 0 | 1 | 18 |
| CAATTATTATAAGAAA | 63 | 233 | 0 | 4 | 11 | 6 | 19 |
| ATCAGACAACTACTAT | 60 | 179 | 0 | 2 | 0 | 0 | 18 |
| ACCTTTGACAATCTTA | 55 | 292 | 0 | 15 | 70 | 67 | 22 |
| ATTTGAAGTCACCTAA | 55 | 258 | 0 | 11 | 36 | 31 | 20 |
| CATGCAAATTACATAT | 54 | 204 | 0 | 3 | 0 | 0 | 18 |
| GAATTTCACAGTATTC | 54 | 260 | 0 | 12 | 38 | 35 | 22 |
| TTTGTAGAAAACCCAG | 53 | 246 | 0 | 7 | 24 | 18 | 20 |
| ATGCTGCAATCGTGCT | 24 | 139 | 0 | 2 | 0 | 0 | 17 |
| ATTTCCTAATATTACA | 24 | 92 | 0 | 1 | 0 | 0 | 17 |
| CTCTATCATTATTGGT | 24 | 121 | 0 | 1 | 0 | 0 | 17 |
| TGTTTTATTCTCTACA | 24 | 199 | 0 | 3 | 0 | 1 | 19 |
| CAGTACATCGATATCG | 23 | 119 | 0 | 2 | 0 | 0 | 17 |
| TAATGGTGACTTTTTG | 23 | 92 | 0 | 1 | 0 | 0 | 17 |
| CAACCATAAAACCAGT | 22 | 105 | 0 | 1 | 0 | 0 | 17 |
| AGTTATTTGACTCCTG | 21 | 97 | 0 | 1 | 0 | 1 | 18 |
| ATAAAGGAGTTGCACC | 19 | 218 | 0 | 5 | 0 | 0 | 18 |
| **Literature** | | | | | | | |
| of the | 12565 | 12630 | 12086 | 12325 | 12463 | 12454 | 12563 |
| in the | 6188 | 6242 | 5698 | 5906 | 6075 | 6067 | 6096 |
| and the | 6173 | 6314 | 5770 | 5972 | 6147 | 6139 | 6169 |
| the of | 6015 | 6162 | 5618 | 5834 | 5995 | 5985 | 6014 |
| the lord | 4186 | 4289 | 3745 | 3975 | 4122 | 4114 | 4185 |
| to the | 3465 | 3558 | 3014 | 3217 | 3391 | 3380 | 3464 |
| the and | 2250 | 2413 | 1869 | 2081 | 2246 | 2237 | 2249 |
| all the | 2226 | 2346 | 1802 | 1993 | 2179 | 2170 | 2225 |
| and he | 2169 | 2293 | 1749 | 1937 | 2126 | 2117 | 2168 |
| to be | 2062 | 2121 | 1577 | 1770 | 1954 | 1945 | 2061 |
| very for | 15 | 59 | 0 | 2 | 0 | 0 | 0 |
| and faithful | 14 | 94 | 0 | 3 | 0 | 0 | 0 |
| but found | 9 | 74 | 0 | 2 | 0 | 0 | 0 |
| my speech | 6 | 98 | 0 | 3 | 0 | 0 | 0 |
| of eight | 5 | 66 | 0 | 2 | 0 | 0 | 0 |
| and soul | 4 | 140 | 0 | 6 | 0 | 0 | 0 |
| her prow | 3 | 79 | 0 | 2 | 0 | 0 | 0 |
| usual as | 2 | 56 | 0 | 2 | 0 | 0 | 0 |
| a invitation | 1 | 80 | 0 | 2 | 0 | 0 | 0 |
| angular log | 0 | 146 | 0 | 5 | 0 | 0 | 0 |

## A4 Additional experiments with two-sided confidence intervals

This section describes additional numerical experiments with synthetic data similar to those described in Figures 1 and A3, constructing two-sided instead of one-sided confidence intervals. For simplicity, we focus on one-sided 95% conformalized bootstrap confidence intervals based on the simpler Bonferroni approach described in Section A1.4.2. The performance of these intervals are compared to those of one and two-sided standard bootstrap confidence intervals obtained with the method of [5].

Figure A17 reports on results based on data generated from a Zipf distribution and sketched with the CMS-CU, similarly to Figure 1. Here, all methods achieve the desired 95% marginal coverage level, but the conformal confidence intervals are shorter when the Zipf tail parameter $a$ is larger and hash collisions become rarer, consistently with Figure 1. In this interesting to note that the two-sided conformal confidence intervals are much narrower than their one-sided counterparts when $a$ is small and hash collisions are very common, but this is not true if $a$ is large. The latter is likely a limitation of the specific construction we have adopted, described in Section A1.4.2, which may be too conservative in some cases due to the Bonferroni correction. A suitable implementation of the more sophisticated conditional histogram [4] approach described in Section A1.4.1 should be expected to produce two-sided intervals that are always narrower than their one-sided counterparts. Figure A18 reports on results similar to those in Figure A17, with the only difference that now the data are sketched with the vanilla CMS instead of the CMS-CU.

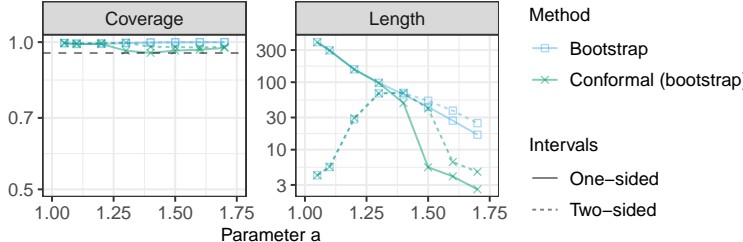

Figure A17: Performance of 95% one-sided and two-sided confidence intervals with data from a Zipf distribution, sketched with the CMS-CU. The results are shown as a function of the Zipf tail parameter $a$. Standard errors would be too mall to be clearly visible in this figure, and are hence omitted. The two dashed curves for the two-sided intervals are nearly indistinguishable from one another for $a < 1.3$. Other details are as in Figure 1.

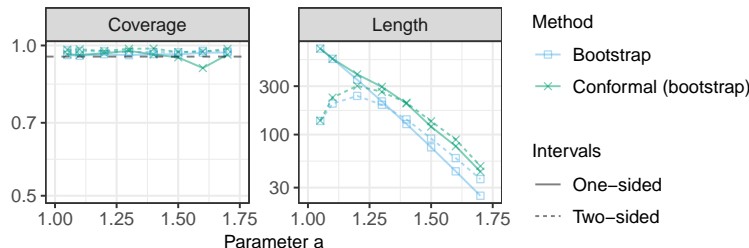

Figure A18: Performance of 95% one-sided and two-sided confidence intervals with data from a Zipf distribution, sketched with the vanilla CMS. The results are shown as a function of the Zipf tail parameter $a$. The two dashed curves for the two-sided intervals are nearly indistinguishable from one another for $a < 1.1$. Other details are as in Figure A17.

Figure A19 reports on results based on data generated from a Pitman-Yor process prior and sketched with the CMS-CU, similarly to Figure A3. As expected, the conformal confidence intervals are narrower than the bootstrap ones. Further, two-sided confidence intervals are much more efficient (narrower) compared to their one-sided counterparts, especially if the Pitman-Yor parameter $\sigma$ is large and the number of hash collisions is high. Figure A20 reports on analogous results obtained with data sketched through the vanilla CMS instead of the CMS-CU.

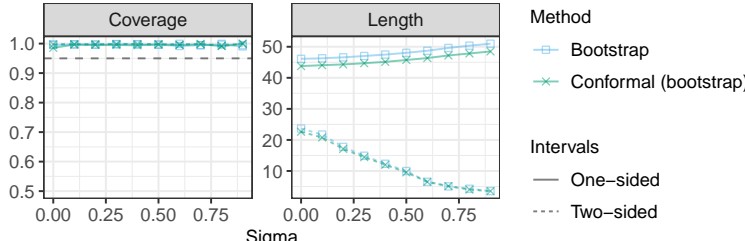

Figure A19: Performance of 95% one-sided and two-sided confidence intervals with data set sampled from the predictive distribution of a Pitman-Yor process and sketched with the CMS-CU. The results are shown as a function of the Pitman-Yor process parameter $\sigma$. The two dashed curves for the two-sided intervals are nearly indistinguishable from one another. Other details are as in Figure A3.

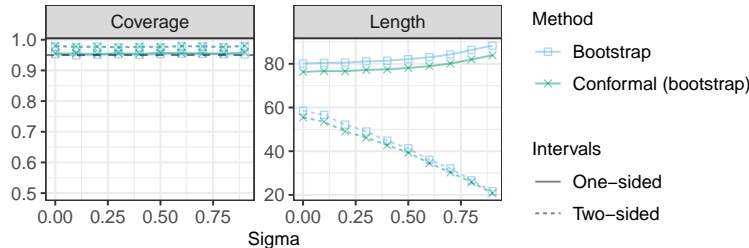

Figure A20: Performance of 95% one-sided and two-sided confidence intervals with data set sampled from the predictive distribution of a Pitman-Yor process and sketched with the vanilla CMS. The results are shown as a function of the Pitman-Yor process parameter $\sigma$. Other details are as in Figure A19.