# OpenReview forum: "Conformal Frequency Estimation with Sketched Data"
_NeurIPS.cc/2022/Conference — NeurIPS 2022 Accept_

### Official Review · Reviewer_vGGR · 2022-07-10

**Rating:** 6
**Confidence:** 4
**Soundness:** 3 good
**Presentation:** 3 good
**Contribution:** 3 good

**Summary:**

The authors propose a method for constructing confidence intervals for counts based on approximate counts generated by a sketching algorithm.  The proposed method involves keeping track of exact counts for a sparse dictionary of items and computing conformity scores based on disparity between the sketched count and true count.  Using these scores, a prediction region is generated for a randomly selected element of the dictionary using conformal prediction.  The method is shown to be competitive on synthetic and real data examples.

**Questions:**

- Is there a reason that you did not consider two-sided prediction intervals?  One would think that it is possible to simply treat all entries of the hash matrix that the item was mapped to as a covariate and then use a standard regression non-conformity score.
- Is there a reason that coverage for conformal prediction is (nearly) 100%?  When the non-conformity scores are distinct, conformal prediction is supposed to have the upper bound $1- \alpha + \frac{1}{n+1}$ on coverage as well.  Does randomizing the scores fix this?

**Limitations:**

The authors are forthright about exchangeability being potentially restrictive for sketching problems, but do not discuss the subtle issues with interpreting the sampled items in the real data examples.

**Strengths And Weaknesses:**

Strengths:
- The proposed method inherits the flexibility of conformal prediction and thus works for any sketching algorithm and for any performance measure i.e. conformity score.
- The construction of the ordered pairs to transform the sketching problem into a supervised learning problem amenable to conformal prediction is novel.
- The fact that it suffices to keep track of $m_0 \ll m$ counts in order to construct a prediction interval for the test object due to the exchangeability assumption is interesting.

Weaknesses:
- One of the main arguments that the authors make in favor of conformal prediction over deterministic approaches is that the deterministic approaches are too conservative.  Yet, the procedures proposed use a deterministic upper bound.  Thus, these prediction intervals are also conservative in the sense that the probability that the true quantity is greater than the upper bound is 0.
- While it is typically a mild assumption, exchangeability is a nontrivial one for sketching problems.  In fact, the 2-gram real data example does not appear to be exchangeable since consecutive 2-grams share a word and thus the joint distribution of 2-grams is not invariant under permutation.  The authors are instead treating the 2-grams as fixed and are sampling IID from them, but then the interpretation of the conformal prediction guarantee is less clear, as the probability statement includes this sampling variation.  In this setup, the prediction interval  covers a sampled frequency instead of the true one.
- Another typically benign feature of conformal prediction that may be problematic in this setting is that the test point is assumed to be exchangeable when combined with the training examples; this means that the one can only construct prediction intervals for randomly chosen items, which is quite restrictive particularly if the dictionary is large.
- The technical contributions of the paper are not substantial.  This is not meant to be a major criticism since the combination of simplicity and generality is what makes conformal prediction useful in a wide variety of areas.
- The procedure becomes trivial if the element of interest appears in the warmup phase or if one is interested in a small number of elements, in which case one may just track these elements following the same reasoning used to justify the warmup phase.
- While the writing is clear, I believe the exposition can be improved in places.  For example, the sketching problem should be first introduced in more generality before the count-min sketch is introduced, since the paper is not solely about CMS or CMS-CU.  Also, it is misleading to refer to the constructed prediction sets as one that “is as short as possible” on line 162 since optimality of prediction sets is a different topic that is not investigated here.

---

> ### Author Response · Authors · 2022-08-02
> **Response to comments by Reviewer vGGR (Part 1, continued below)**
>
> Weakness 1. This comment is due to some misunderstanding which we can resolve by improving the exposition, and we do not think it points to a true weakness in this paper. Indeed, our novel methodology is not at all limited to one-sided intervals. All the key methodological components of our paper, in Section 3.1 and 3.2, are designed to accommodate two-sided intervals; see for example Algorithm 2 (in the appendix) and Theorem 2. The reason why our practical demonstrations, in Section 3.3 and Section 4, focus on one-sided intervals is the same reason why we pay special attention to the CMS and variations thereof: this facilitates the comparison with related prior work, and especially with the classical approach which gives us a probabilistic lower bound and a deterministic upper bound. Further, the exposition of our conformity scores is a little easier to explain for one-sided intervals (Section 3.3), although the two-sided counterparts would not be much longer to explain because they are already available from Sesia and Romano (2021), for example. That being said, we agree that it would also be interesting to compare the performance of our conformal intervals to that of the Bayesian and bootstrap (Ting, 2018) ones in the two-sided case. We would be very happy to add such comparisons to this paper if we are given the opportunity to revise it. Even though did not have time to add the-two sided comparisons during this rebuttal phase, this is a relatively straightforward change which does not require any additional methodological novelty. Therefore, there is no doubt it would only involve a minor revision. All we need to do is to replace the special conformity scores in Section 3.3 with their more general two-sided counterpart developed by Sesia and Romano (2021), and then re-run the experiments. In hindsight we wish we had already done so in the submitted manuscript, so thank you for bringing this opportunity for improvement to our attention. Finally, note that there is no reason to expect any surprises from these additional two-sided experiments. Our one-sided intervals are already much shorter than the classical ones, and they will become even shorter if we allow them to be two-sided. The Bayesian and bootstrap intervals have no more advantage against ours in the two-sided case than they have in the one-sided case; in fact, there is no substantial difference in how any of these methods operate under the one-sided vs. two-sided framework.
>
> Weakness 2. This comment is also due to some misunderstanding which we can resolve by improving the exposition, and we do not think it points to a true weakness in this paper.
> First, it is clear that the 2-gram data (or the k-mer data) would not be exchangeable if we were to process (e.g., sketch and query) the 2-grams in the same order in which they appear in the original natural language document. However, this is not what we do in our experiments (you can verify this by looking at the submitted code) and it is not how we would like practitioners to apply our method. Unfortunately, it appears that we accidentally forgot to explain this point in the paper. We are grateful for the opportunity to clarify. What we do in our 2-gram experiments is to sample i.i.d. 2-grams from the collection of all possible 2-grams in our data set. This data-generation mechanism automatically satisfies the exchangeability assumption, and it would not be unreasonable to approximately follow the same idea in practice. For example, one can always process a data set stored on hard drive in a random order; this approach may be slower than sequential reading, but it is feasible and, combined with sketching, it still allows one to be memory-efficient. In general, we should make it even more explicit that practitioners should only apply our method to data that are either exchangeable to begin with, or which can be made exchangeable by suitable randomization as we do in our experiments. Of course, randomizing the data is not always feasible (e.g., it cannot be done in the case of an online data stream), but that is why we clearly state that our method is not going to be a panacea for all possible applications.
> Second, the other existing methods (Bayesian and bootstrap) also assume i.i.d. data, so this limitation is not specific to our paper. The only existing approach that makes no i.i.d. assumptions is the classical one, but in that case the price to pay is that the bounds are always impractically wide.
> Third, we should clarify that our exchangeability assumption can be relaxed insofar as the query points are concerned, but this is discussed in more detail in our next comment.

---

> > ### Author Response · Authors · 2022-08-02
> > **Response to comments by Reviewer vGGR (Part 2, continued below)**
> >
> > Weakness 3. Again, this comment is also due to some misunderstanding which we can resolve by improving the exposition, and we do not think it points to a true weakness in this paper.
> > First, the assumption that the test point is exchangeable when combined with the training examples generally never is a “benign” assumption. It is a very useful assumption, which has allowed many successful applications of conformal inference and has not prevented the rapid growth of this field, but the literature is very well aware of the delicacy of exchangeability. A lot of effort has been dedicated to relax it as much as possible; see for example Vovk et al. (2005), Tibshirani et al. (2019), Barber et al. (2022). The truth is that we have also taken rigorous and effective measures to mitigate our reliance on such assumption in this paper, but perhaps we did not explain this important point as well as we could have. We are very grateful to have the opportunity to clarify this.
> > In hindsight, we believe the second part of Section 3.1, starting from line 152, should have been explained more carefully and placed into its own sub-section, with a sub-title such as “Relaxing the exchangeability assumption with frequency-conditional coverage”. Our method is specifically designed to control the notion of frequency-conditional coverage defined in Equation (8). This notion of coverage is stronger than the standard marginal coverage in Equation (7), and it is specifically intended to deal with the fact that it may be preferable not to treat the test point as exchangeable with all the training data. Controlling the stronger coverage in Equation (8), as our method provably does, is essentially equivalent to (partially) relaxing the exchangeability assumption. More precisely, our method allows us to to achieve provably valid coverage even conditional on a query being relatively rare. In other words, this means that our coverage guarantee still holds even if some covariate shift occurs in the test set, causing queries which were previously rare in the training data to suddenly become more common (or the other way around). In particular, this is precisely why it is not true that we “can only construct prediction intervals for randomly chosen items, which is quite restrictive particularly if the dictionary is large.” That would be accurate if we could only control Equation (7), but we can control Equation (8). In fact, this is a significant relaxation of the general exchangeability assumption which very intuitive and useful in our frequency estimation problem. We have now realized that the value of this important but somewhat subtle component of our contribution might have been missed by a large audience because it was not explained very carefully. We hope we have answered your question to satisfaction, and that you will give us the chance to incorporate these clarifications into the paper. Of course, this is not to say that we have completely removed the exchangeability assumption. There is more work to do, such as to address the interesting open question mentioned in Section 5. However, we are leaving that to follow-up work because it is a pretty searching issue by itself.

---

> > > ### Author Response · Authors · 2022-08-02
> > > **Response to comments by Reviewer vGGR (Part 3, last)**
> > >
> > > Weakness 4. We are not trying to claim this paper introduces ground-breaking theoretical advances, because it does not, but perhaps it is not fair either to say our technical contributions are not substantial enough. There has been a lot of recent activity in conformal inference, while other unrelated papers have started to consider the problem of tightening the error bounds for the CMS by adopting randomized data perspectives; e.g., Ting (2018) and Cai et al. (2018). Yet, our rigorous mathematical formulation of the sketching problem into a conformal prediction framework had not been suggested by anyone else before. While we do build upon existing conformal prediction ideas, this is hardly a standard application of conformal prediction. Clearly, the connection was not sufficiently obvious to spur anyone else before to make it. Another reviewer has praised this novelty: “The paper's biggest strength is the sheer novelty of the combination of conformal prediction and data sketching. To my knowledge, the combination is completely new.” Further, our stronger notion of coverage defined in Equation (8) is also novel. This was inspired by prior work on Mondrian conformal prediction (Vovk et al., 2005), but it is original insofar as it specifically addresses your previous comment about the issue of exchangeability in the sketching problem. We understand that you might have not previously fully appreciated the importance in the context of sketching of the stronger coverage defined in Equation (8), which our method can provably guarantee. This is our fault: we didn’t explain it as well as we should have. However, given that this issue has been clarified, would you re-assess the technical contributions of the paper?
> > >
> > > Weakness 5. It is true that our procedure becomes trivial if the element of interest appears in the warmup phase, or if one is interested in a small number of elements, in which case one may just track these elements following the same reasoning used to justify the warmup phase. However, how is this a weakness of our method? There is definitely some weakness in our exposition, as we should have mentioned this explicitly. We will do so if given the opportunity to revise. However, this fact indicates a strength of our method, not a weakness: for some queries we can simply get perfectly tight frequency bounds for free. Note that we did not take advantage of this cool fact in our numerical experiments because it would have not seemed 100% fair towards the other methods, which do not have access to the warm-up data. However, if we did take advantage of this, as one should do in practice, we would see that our conformal prediction sets would become even better! Thank your for reminding us to mention this!
> > >
> > > Weakness 6. We are not going to argue with you on this final point: our exposition can be improved. You have already pointed out several opportunities for clarification in the above comments, and we are very grateful for that. We are convinced these improvements will make the paper much more broadly accessible, but they do not require major methodological or conceptual changes. Therefore, we are completely confident that we could take care of them before a camera-ready version of the paper is due, if we are given the opportunity to revise our submission.
> > >
> > > Question 1. Right, it is definitely possible (and easy) to apply our method to construct two-sided intervals. We have already discussed this in our answer to your comment above. In short, the reason why our applications focused on one-sided interval was to ensure consistency with the classical approach, but we can easily add more implementation details and empirical results for the two-sided case.
> > >
> > > Question 2. You are right: the coverage should always be nearly tight in theory if the conformity scores  have a continuous distribution. The conformity scores we use here are not continuous, and that makes sense because the problem is intrinsically discrete. However, it is also true that should be able to obtain even shorter valid prediction intervals if we make the conformity scores continuous by adding some randomization; see for example Romano et al. (2020). We did not randomize the conformity scores because we thought it would unnecessarily complicate the notation, but we would be happy to explain that extension in the revised paper if you think it can be useful to do so.
> > >
> > > Limitations. We hope our previous comments clarified that indeed our paper does consider quite carefully the subtle issues involved with the data randomness and the exchangeability of the query points, even though our exposition did not clearly reflect it. We will make sure to improve the exposition accordingly if given the opportunity to do so.

---

> ### Author Response · Authors · 2022-08-09
> **Friendly reminder about the upcoming end of the author discussion period**
>
> Dear Reviewer vGGR,
>
> Thank you again for your careful read and detailed comments about our manuscript. We have taken your feedback very seriously and we have as a result identified several areas in which our exposition could be made clearer. We feel that our paper will improve significantly as a result of your feedback, and we are grateful for that. We were just wondering whether you could kindly let us know whether you are satisfied by our responses, or whether you have any remaining questions which we might be able to address in the final few hours before this author discussion period ends.
>
> Sincerely,
> The authors

---

> > ### Comment · Reviewer_vGGR · 2022-08-09
> > **Response to Authors**
> >
> > We thank the authors for providing a detailed response and addressing some of the concerns that were raised.  In response to your comments, I am willing to increase the score for the paper slightly, but would like to clarify that the vast majority of the shortcomings that I discussed were NOT misunderstandings on my part.  The paper is interesting, but the typically mild assumption of exchangeability needed for conformal prediction happens to be stronger for the sketching problem considered in the paper.
> >
> > - Weakness #1: While the general approach does not require one-sided confidence intervals, the only nonconformity scores defined (and studied in the simulation and real-data examples) in the paper (in particular, Section 3.3) were one-sided with the deterministic upper bound.  This is why I also ask in the question section why the two-sided intervals were not considered explictly.  One would think that the one-sided prediction intervals considered have an artificially high upper bound and the lower bound attempts to compensate for this, which may be undesirable.  One may argue that this is more of a writing issue than content, but the nonconformity score is undoubtedly part of the method and this was the only one considered.
> > - Weakness #2:  I explicitly state in my review that the 2-grams are treated as fixed but are sampled from in an iid manner.  While this leads to “validity,” what is now covered is not the underlying frequency but a sampled frequency and that the probability measure involves this sampling.
> > - Weakness #3:  I am aware of the conditional validity literature, but the stated approach is not a panacea.  One has to partition the examples a priori.  In other words, one has to decide what is the collection of rare examples for which one wants conditional coverage for before seeing the data.  One may not use any other additional information to choose the item of interest.  Again, this is not a major weakness, but was simply an observation regarding the limitations of exchangeability in this setting.  Exchangeability is satisfied in common idealized setups, so it is more or less “benign” or harmless, although people have studied cases when this assumption is violated in certain ways.
> > - Weakness #4: I also explicitly stated in the review that the extent of the technical contribution is NOT a major weakness.  In the strengths, I actually pointed to the novelty of the construction of the conformity scores.
> > - Weakness #5: This was more of an observation about the difficulty of the problem when one is willing to assume a warmup phase - in certain cases of interest, conformal prediction is not even needed.

---

> > > ### Author Response · Authors · 2022-08-09
> > > **Re: Response to Authors**
> > >
> > > Dear Reviewer vGGR,
> > >
> > > Thank you for taking the time to read our rather long response and for continuing the discussion. Please let us clarify that we did not mean to suggest any misunderstanding might be due to any fault on your side. By contrast, we were very pleased to see you read our paper carefully. What we meant to convey with the term "misunderstanding" was that some of the concerns you raise (e.g., the delicacy of exchangeability assumptions) are not problems of which we are unaware, or which we want to hide. The "misunderstandings" to which we referred are due to our less-than-perfect exposition, which we are however fully prepared to correct in the next round of revision or in the camera-ready version, if the paper is accepted. Specifically, we will clarify the following points (we can call them that if we don't like the word "misunderstanding"):
> > >
> > >  - Our method is not specifically tied to the one-sided conformity scores currently employed. It is completely straightforward to apply our method with existing two-sided conformity scores such as those found in Sesia and Romano (2022), and all our theoretical results will remain equally valid. We will write down explicitly how to compute the two-sided scores and then we will repeat the experiments with these scores. We agree this will be a nice extension, and we are grateful for the suggestion, but it really isn't a big deal to implement it. The technical and conceptual novelty of our paper is the recasting of the sketching problem into conformal inference framework, not the choice of conformity scores.
> > >
> > >  - The issue of exchangeability is indeed more subtle in the sketching context than in other applications of conformal prediction. We had mentioned this point in some places, but we agree it needs to be emphasized and explained even more clearly. In particular, we will better explain that the frequency-conditional methodology and the frequency-conditional performance metrics presented in our paper are specifically designed to mitigate the exchangeability issues. This is not to say we have completely resolved the exchangeability issues: to the contrary, we are already proposing in the discussion one interesting but more technically challenging way to further reduce our reliance on exchangeability in the future. However, we do not think the exchangeability assumptions are sufficiently problematic to justify discarding conformal inference as a viable and potentially very promising framework for uncertainty estimation with sketched data. To the contrary, exchangeability is a subtle and important topic that needs to be discussed carefully, and which will need to be researched further in this context. We would like to think our paper takes a first meaningful step in that direction, while providing motivation and ideas for further work.
> > >
> > >  > One has to partition the examples a priori. In other words, one has to decide what is the collection of rare examples for which one wants conditional coverage for before seeing the data.
> > >
> > > Well, let's just clarify we only need to decide a priori what range of frequency values qualifies as "rare", not which specific examples are rare. That being said, it's true that we cannot achieve the "full-conditional" coverage that one would ideally hope for, but this is a fundamentally tough problem. It's not very reasonable, at least not at this point of the methodological development, to demand full conditional coverage in the generality that we work in. The only other existing approaches to solve our problem are limited to deal with specific (linear) sketching algorithms or must operate under an extremely conservative "adverserial data" framework.
> > >
> > > >  I also explicitly stated in the review that the extent of the technical contribution is NOT a major weakness. In the strengths, I actually pointed to the novelty of the construction of the conformity scores.
> > >
> > > Thank you!
> > >
> > > > This was more of an observation about the difficulty of the problem when one is willing to assume a warmup phase - in certain cases of interest, conformal prediction is not even needed.
> > > Thank you for pointing this out. We just wanted to say we also thought about it, and it was a distraction on our side not to mention it. We will fix it!
> > >
> > > Thank you again for the very helpful comments!

---

### Official Review · Reviewer_gvFV · 2022-07-11

**Rating:** 4
**Confidence:** 4
**Soundness:** 4 excellent
**Presentation:** 3 good
**Contribution:** 2 fair

**Summary:**

The paper proposes a conformal method that, given a sketching function $\phi$, computes a confidence interval for the frequency of a random query sampled exchangeably from the data.

**Questions:**

1. A coverage guarantee with respect to the sketching randomness feels natural for working with sketched data. By contrast, it is still not obvious to me that a coverage guarantee with respect to the draw of data points can be useful in many situations. Could you provide additional examples in which the latter guarantee is of equal or greater interest? The more concrete the examples, the better. Examples from prior applications would be excellent.

2. It looks like the proposed method and the comparison methods all have different theoretical guarantees. For example, for the classical method, the probabilistic statements are made with respect to the randomness of the hash functions, whereas for the proposed method, they are with respect to the sampling of the new query point and the sketched data. (I am not familiar with other comparison methods.) Should we exercise more caution in interpreting the experimental results if this is the case? To what extent are the proposed method and the comparison methods truly comparable?

3. On a related note, how should I interpret these different confidence intervals? I know how to interpret the classical bound when I have an approximate answer based on the sketched data, and I want to relate it to the exact answer based on the un-sketched data. However, I am struggling to find an interpretation for the conformal interval.

4. Minor
- Figure 1: too mall -> too small
- The captions for figures and tables could be made more informative. It was a bit annoying to keep flipping back and forth between an earlier figure and a later one to understand what the latter was about.

**Limitations:**

The authors appear to be aware of the limitations of their method (cf. Section 5). However, I think that these limitations deserve a more extensive treatment.

**Strengths And Weaknesses:**

The paper's biggest strength is the sheer novelty of the combination of conformal prediction and data sketching. To my knowledge, the combination is completely new.

Unfortunately, the method has at least two significant limitations. First, it is unclear whether the type of theoretical guarantee that the method can provide is of interest for working with sketched data since the probability of coverage is with respect to the randomness of the query and not of the sketching. Second, for this guarantee to hold, the random queries must be exchangeable.

---

> ### Author Response · Authors · 2022-08-02
> **Response to comments by Reviewer gvFV (Part 1, continued below)**
>
> Question 1. We believe your statement that "A coverage guarantee with respect to the sketching randomness feels natural for working with sketched data." can be debated.
>
> True, most of the literature on sketching has taken a worst-case view and treats only the sketching function as random. This historical perspective is acknowledged in the introduction, and we do not deny it may sometimes remain preferable. Yet, why should that worst-case view feel generally more natural to machine learning experts? Broadly speaking, one could say the shared goal of machine learning and statistics is to “learn from data” something meaningful about a distribution (e.g., how to make predictions, or how to estimate an unknown parameter). Intuitively, in order to learn one must typically assume at least some sort of randomness in the data, and it is not uncommon to go as far as to assume i.i.d. samples. Then, even more assumptions (e.g., parametric models) tend to be needed for more detailed theoretical studies. Thus, from the perspective of many (or perhaps even most) NeurIPS readers, it should feel counter-intuitive to look at our sketched frequency estimation problem from a worst-case perspective, especially if an alternative view is offered.
> The results in our paper show that the deterministic approach always leads to very wide intervals, precisely because it cannot learn from data. This limitation was already known; for example, Ting (2018) writes: “Although the Count-Min sketch has been useful for estimating counts, the problem of returning a practical error bound has not been addressed before”. That is because the classical bounds are not very practical. Now, it is true that sometimes one has no choice but to take a very pessimistic view (e.g., if the data collection mechanism may be adverserial), and in those cases one should stick with classical sketching. However, this is a machine learning conference, so it makes sense for us to try to learn something from the data.
>
> Next, it is worth repeating that our paper is not the first one to look at sketching from a machine-learning perspective in which the data are treated as random. Two related lines of work precede us. On the one hand, Ting (2018) took a frequentist perspective similar to ours but focused only on the linear CMS. Ting’s method is acknowledged in Cormode and Yi’s 2020 book “Small summaries for big data”. (Note: Graham Cormode is one of the fathers of the CMS algorithm). On the other hand, there is a growing Bayesian line of work which started from Cai et al. in NeurIPS (2018); this work also looks at sketching from a “random data” perspective. The Bayesian approach computes a posterior distribution, but it has the downside of making modeling assumptions which can become problematic if they are mis-specified (Figure 1). The Bayesian approach is also limited to the linear CMS. This is where our method comes in: we can take the random-data perspective of Ting (2018) and Cai et al. (2018) one long step further, by completely removing all modeling assumptions and all restrictions on the form of the sketching algorithm.
>
> Finally, classical deterministic bounds only exist for few relatively simple sketching algorithms. Even though the applications in this paper focus on the CMS and variations thereof for simplicity, our method is much more general: it can be applied to any arbitrarily complex and possibly unknown sketching algorithm. There is simply no existing alternative which can assess sketching uncertainty in a practical way and under such generality.
> In conclusion, we agree that the introduction could be improved for clarity. In part, our conciseness was due to the space limitations. Fortunately, we can distill this discussion into the extra page allowed if our paper is accepted.
>
> Regarding your question about applications, we start by referring to Cai et al. (2018). Next, the demonstrations in Sections 4.2-4.3 provide more concrete examples. Our first example is about counting k-mers from DNA data, and the second one is about counting n-grams in text. These are classical sketching applications; see Cormode and Yi’s 2020 book. Yet, the potential impacts of our work do not stop there. As mentioned above, we can deal with any sketching algorithm. This flexibility is very valuable because it is plausible that sketching algorithms will become more pervasive and diverse in the future. First, we have lots of “big data” that are expensive to process and transfer; sketching can make them more efficient to deal with. Second, there are privacy and fairness considerations creating incentives to work with different types of “sketched” data instead of raw sensitive data (Melis et al., 2016; Corrigan-Gibbs and Boneh, 2017). The conformal inference ideas discussed in this paper may thus become even more relevant to sketching in the near future, including in the rapidly growing field of federated learning (Li et al., 2019; Rothchild et al., 2020; He et al., 2020).

---

> > ### Author Response · Authors · 2022-08-02
> > **Response to comments by Reviewer gvFV (Part 2, continued below)**
> >
> > Question 2. As discussed above, the classical method can only make probabilistic statements with respect to the randomness of the hash functions; this inevitably results in extremely wide confidence intervals. Unfortunately, there is not much we can do to fix this issue, other than developing the alternative method that we develop in this paper. The point of Ting (2018), Cai et al. (2018), and of our paper is precisely this: the classical confidence intervals for sketching are often too wide to be of much practical use. As far as we know, overcoming this limitation of the classical bounds requires taking a “learning from the data” perspective, and so that is what we do. Therefore, in a certain sense it is true that our confidence intervals are not designed to solve quite the same problem as the classical confidence intervals, and therefore the comparison should be interpreted carefully. However, this is hardly a weakness of our paper. If we had not compared our intervals to the classical ones, some reviewer would have almost certainly asked us to do it. In fact, it is very important to compare our intervals to the classical ones, because such comparison demonstrates very clearly how the classical intervals are often extremely wide. Next, regarding the comparison between our intervals and those of Ting (2018), they are actually quite similar in the case of the linear CMS, as we discuss in the paper. Finally, regarding the comparison between the frequentist intervals vs. the Bayesian ones, this is also a fair comparison because we carry out repeated experiments. In other words, it makes perfect sense to evaluate the performance of a Bayesian method over repeated experiments according to frequentist metrics (average coverage and length). Such comparison would be more problematic if our paper took a Bayesian perspective and did not repeat multiple experiments, but than that is not what we do here.

---

> > > ### Author Response · Authors · 2022-08-02
> > > **Response to comments by Reviewer gvFV (Part 3, last)**
> > >
> > > Question 3. The interpretation of our conformal confidence intervals satisfying Equation 7 is the following. Suppose we were to repeat infinitely many times the multi-step experiment consisting of (1) sampling data from the underlying data-generating distribution, (2) sketching it, (3) querying a random object sampled from the same data-generating distribution, and (3) constructing a conformal prediction interval for the true frequency of the queried object among the sketched data. Then, 90% of those infinitely many conformal prediction intervals would contain the corresponding true frequencies. In other words, the marginal probability (with respect to all randomness in this experiment, except the sketching randomness which can be safely fixed) that the true frequency of a queried object is in its corresponding conformal prediction interval is 90%. This is much weaker than the classical probabilistic guarantee, but we say so very explicitly. The point is that the classical probabilistic guarantee is often so strong as to become very impractical to achieve.
> > >
> > > It is worth pointing out that our method can also produce conformal confidence intervals satisfying a stronger notion of coverage: Equation 8. In the paper, we refer to this notion of coverage as “frequency-conditional” coverage, and we evaluate it in the fashion of Figure 2. “Frequency-conditional” conformal coverage can be interpreted as follows. Suppose we were to repeat infinitely many times the multi-step experiment consisting of (1) sampling data from the underlying data-generating distribution, (2) sketching it, (3) querying a random object sampled from the same data-generating distribution, and (3) constructing a conformal prediction interval for the true frequency of the queried object among the sketched data. Suppose also that we discard the results of all experiments in which the random queried object falls outside of the desired frequency bin. Then, 90% of that infinite subset of remaining conformal prediction intervals would contain the corresponding true frequencies. In other words, the conditional probability that the true frequency of a queried object is in its corresponding conformal prediction interval given that the query is assigned to a specific confidence bin is 90%. This is still weaker than the classical probabilistic guarantee, but it is stronger than the marginal coverage discussed above.
> > >
> > > It is also useful to recall that the guarantee of Ting (2018) is similar to our frequency-conditional coverage with bins of size 1, as mentioned in our paper. Therefore, Ting’s guarantee is somewhat stronger than ours in most cases, but ours has the advantage of being applicable beyond the linear CMS. The Bayesian guarantee is a bit different, because it treats the query as fixed but models the data as random with a specific data-generating model. However, if the Bayesian prior is well-specified, and the experiment is repeated many times in the frequentist sense explained above, then the Bayesian intervals will still satisfy our notion of coverage. Our experiments confirm empirically that this is indeed the case, although they also highlight how the Bayesian solution fails if the prior is mis-specified. We will be happy to improve the paper  exposition in light of this discussion. Further, the Bayesian approach is also theoretically limited to the linear CMS, at least for now.

---

### Official Review · Reviewer_HNPK · 2022-07-17

**Rating:** 7
**Confidence:** 2
**Soundness:** 3 good
**Presentation:** 3 good
**Contribution:** 3 good

**Summary:**

This manuscript develops conformalized sketching: a method using conformal prediction to construct confidence intervals for frequency queries based on sketched data. The method works at any desired level, unlike the standard version of count-min sketch, and also incorporates conservative updates. Theoretical results and simulations (with both synthetic and real-life datasets) are provided which establish validity and showcase the appeal of the proposed method.


UPDATE: after reading the author's response, I have increased my score to a 7 (accept).

**Questions:**

Further discussions about the length of the confidence intervals, and also performance when the exchangeability assumption does not hold would be helpful. For example, why are the confidence intervals for conformalized sketching so short in the simulation results?

**Limitations:**

See questions and the weakness comment. Potential negative societal impact not applicable.

**Strengths And Weaknesses:**

Strengths: the paper is well written and appears to be up to date with the literature. A overview of conformal prediction is provided and the results are self-contained.

Weakness: more theoretical results would be helpful. For example, some theoretical results concerning the length of the confidence intervals, and also performance when the exchangeability assumption does not hold would be helpful.

---

> ### Author Response · Authors · 2022-08-02
> **Response to comments by Reviewer HNPK**
>
> Studying theoretically the length of the confidence intervals. This is a good question, but unfortunately it is not an easy one to answer. The challenge is that the length of our confidence intervals will generally depend on the unknown data distribution, on the sample size, on the specifics of the sketching algorithm, and on the chosen form of the conformity scores. These are a lot of complex moving parts. In fact, there are so many complex moving parts that it should already feel quite remarkable that we can get rigorous coverage guarantees for our algorithm. That being said, there are several works in the literature which study theoretically the length of conformal prediction intervals in some settings; see for example Lei et al. (2018), Sesia and Candes (2020), or Sesia and Romano (2021). However, those theories require so many additional assumptions that their results are only easy to interpret insofar as they are utilized to compare the relative efficiency of different conformal inference techniques. Therefore, even if we went through the rather time-consuming exercise of carrying out similar theoretical analyses in the context of sketching, it is not so clear what new insight we could possibly gain. At the moment, our method is the only one of its kind in the context of frequency confidence interval estimation from arbitrarily sketched data, and it offers technically different guarantees compared to other types of approach. Thus, we remain convinced that it is more meaningful to compare the length of each method’s confidence intervals empirically, at least within the scope of this paper.
>
>
> Exhangeability. This is also a very good question and it is related to a similar comment by Reviewer gvFV. In truth, we are already evaluating the performance of our method from a perspective that goes well beyond full exchangeability. For example, Figure 2 summarizes the performance of our confidence intervals separately for rarer and more frequent queries. Stratifying queries by their training population frequency means that we are moving beyond excheangeability. Let us elaborate on that. The left-hand-side of Figure 2 tells us that our confidence intervals would have the desired coverage even all the test queries involved objects with relatively low frequency in the training set. Therefore, we are already doing at least some of what you are suggesting we should do. Of course, there are also other possible ways in which exchangeability may be violated, but a shift in the training frequency of the test queries is arguably the foremost concern that one should have in the context of sketching. Fortunately, our whole method is precisely designed to deal with that! To highlight that, we would like to refer you back to the second part of Section 3.1, the frequency-conditional notion of coverage in Equation (8), and the details of Algorithm 2. See also our response to a similar comment by Reviewer gvFV for further details on how we are dealing with the limitations of exchangeability.
> As we also admit in our response to Reviewer gvFV, not everything in our paper was as clearly explained as it could have been. In particular, the subtle but crucial connection between the limitations of the exchangeability assumption and Equation (8) was perhaps not very accessible, and neither was the discussion of our solution for moving (partially) beyond exchangeability by guaranteeing the stronger frequency-conditional coverage defined in Equation (8). We would be very happy to incorporate these discussions and clarify the exposition if given the opportunity to revise the paper.
>
>
> Why are our confidence intervals so short in the simulation results? For starters, our intervals are generally much shorter than the classical ones because the latter are extremely conservative, to the point of being often impractical. This is a known issue; for example, Ting (2018) writes: “Although the Count-Min sketch has been useful for estimating counts, the problem of returning a practical error bound has not been addressed before”. By contrast, our intervals are not too much shorter than those obtained with the method of Ting (2018) or with the Bayesian approach. Partly, they can be a little shorter because our method can in principle take advantage of the specific data structure induced by any sketching algorithm, while the other two methods are designed for the linear CMS. Therefore, it is intuitive that we can achieve some improvements when we deal with data sketched through more powerful non-linear versions of the CMS. Partly, our intervals can be a little shorter than those of Ting (2018) simply because we guarantee a somewhat weaker form of coverage. As mentioned in the paper, Ting (2018) controls the strongest possible version of our frequency-conditional coverage in Equation (8), while we typically have to work with wider bins (weaker coverage). Of course, our advantage is that we are not theoretically limited to the linear CMS.

---

### Author Response · Authors · 2022-08-02
**Response to reviews**

Dear Reviewers HNPK, gvFV, and vGGR,

Thank you for reading our paper carefully and for providing many insightful comments. We have answered your questions point-by-point below, and we thank you in advance for taking the time to review our responses.

We have learnt from this first round of discussion that some parts of our paper could have been explained more clearly. We hope you will give us the opportunity to improve the exposition based on this discussion. We also hope that our responses address your concerns and clarify all possible sources of confusion. Of course, we would be very happy to continue the discussion if you have any remaining/follow-up comments or questions!

Thank you!

The anonymous authors

---

### Meta-Review · Area_Chair_4C7m · 2022-08-26

**Recommendation:** Accept
**Confidence:** Less certain

**Metareview:**

The paper proposes a method based on conformal inference in order to obtain confidence intervals for the frequencies of queried objects in very large data sets, based on sketched data. The applicability of the method relies solely on the exchangeability assumption for the data, not on the sketching procedure nor on the data distribution, and is therefore very general, as emphasized by all reviewers.

The reviewers have done a great job and this should (and has been) acknowledged by the authors. There has been some objections concerning the applicability of the main assumption (exchangeability), the meaningfulness of the experimental comparison with prior work and the interpretation of the resulting plots, or on the amount of theoretical content of the paper. But the post-review discussion appears to have been very active and fruitful. It overall gives me the impression that the authors took very seriously the comments and will improve the manuscript accordingly, and that many objections could be answered by a more appropriate exposition.  Given that this paper lies of the edge of the acceptance threshold, this improvement is very important, as the reviewers concerns (which have some strong overlap) will otherwise be probably be shared by the wider audience of readers. This is especially true given the statistics flavor of the paper which does not target the main NeurIPS audience, implying that an even greater effort has to be put on the presentation. Very detailed, and I find meaningful from a layman perspective, answers have been provided by the authors, and not all their content will fit in the additional page. There is thus a important work of selection and re-writing ahead of the authors before publication.

**Award:**

No

---

### Decision · Program_Chairs · 2022-09-14

Accept